# Senescent response in inner annulus fibrosus cells in response to TNFα, H₂O₂, and TNFα-induced nucleus pulposus senescent secretome

**Aaryn Montgomery-Song**[1], **Sajjad Ashraf**[2], **Paul Santerre**[3], **Rita Kandel**[1,2,3]*

1 Laboratory Medicine and Pathobiology, University of Toronto, Toronto, Canada, 2 Pathology and Laboratory Medicine, Mt. Sinai Hospital and Lunenfeld-Tanenbaum Research Institute, Toronto, Canada, 3 Biomedical Engineering, University of Toronto, Toronto, Canada

* rita.kandel@sinaihealth.ca

**Data Availability Statement:** All relevant data are within the paper and its Supporting information files.

## Abstract

Senescence, particularly in the nucleus pulposus (NP) cells, has been implicated in the pathogenesis of disc degeneration, however, the mechanism(s) of annulus fibrosus (AF) cell senescence is still not well understood. Both TNFα and H₂O₂, have been implicated as contributors to the senescence pathways, and their levels are increased in degenerated discs when compared to healthy discs. Thus, the objective of this study is to identify factor (s) that induces inner AF (iAF) cell senescence. Under TNFα exposure, at a concentration previously shown to induce senescence in NP cells, bovine iAF cells did not undergo senescence, indicated by their ability to continue to proliferate as demonstrated by Ki67 staining and growth curves and lack of expression of the senescent markers, p16 and p21. The lack of senescent response occurred even though iAF express higher levels of TNFR1 than NP cells. Interestingly, iAF cells showed no increase in intracellular ROS or secreted H₂O₂ in response to TNFα which contrasted to NP cells that did. Following TNFα treatment, only iAF cells had increased expression of the superoxide scavengers *SOD1* and *SOD2* whereas NP cells had increased *NOX4* gene expression, an enzyme that can generate H₂O₂. Treating iAF cells with low dose H₂O₂ (50 μM) induced senescence, however unlike TNFα, H₂O₂ did not induce degenerative-like changes as there was no difference in *COL2*, *ACAN*, *MMP13*, or *IL6* gene expression or number of COL2 and ACAN immunopositive cells compared to untreated controls. The latter result suggests that iAF cells may have distinct degenerative and senescent phenotypes. To evaluate paracrine signalling by senescent NP cells, iAF and TNFα-treated NP cells were co-cultured. In contact co-culture the NP cells induced iAF senescence. Thus, senescent NP cells may secrete soluble factors that induce degenerative and senescent changes within the iAF. This may contribute to a positive feedback loop of disc degeneration. It is possible these factors may include H₂O₂ and cytokines (such as TNFα). Further studies will investigate if human disc cells respond similarly.

**Funding:** This work was funded by the Canadian Institute of Health Research [#142325 and #479529 to RK and PS]. The funders had no role in study design, data collection and analysis, decision to publish, or preparation of the manuscript.

**Competing interests:** The authors have declared that no competing interests exist.

## Introduction

Intervertebral disc (IVD) degeneration has a lifetime prevalence of up to 80% and is associated with the development of back pain [1, 2]. IVD degeneration has a substantial worldwide burden, estimated to affect 266 million individuals annually, leading to significant health care costs and lost wages [3–7]. Despite the prevalence of IVD degeneration, the etiology and pathogenesis of degeneration is still poorly understood. More recently, an association between the pathological changes in the IVD and the presence of senescent cells has been identified [8–10]. Previous studies have found significantly more senescent cells in tissues from human herniated discs [11] than healthy discs. Similarly, aged or degenerative IVDs in mice [12] and rats [13] accumulate significantly more senescent cells than healthy discs. Interestingly, a p16 knockout mouse, a protein identified as a key driver of the senescence program, leads to amelioration of specific markers of IVD degeneration [12, 14]. Further, treatment with senolytic drugs have been shown to reduce IVD degeneration severity [15–17]. These data suggest that cellular senescence plays a role in the pathophysiology of IVD degeneration.

The intervertebral disc is composed of a nucleus pulposus surrounded by annulus fibrosus which can be divided further into an inner and outer zone based on the composition. The inner annulus is integrated with the nucleus pulposus. There is a high degree of variability in the reported senescence rate within the tissues of the IVD in humans, ranging from 13–86% in the NP [11, 13, 14, 18] and 5–86% in the AF [13, 19, 20]. This variation likely reflects the method of senescence identification. One of the most well studied inducers of senescence in NP cells is TNFα [9, 21], however the mechanism through which TNFα induces senescence is still not fully delineated. Studies have implicated PI3K-Akt [21], pSTAT3 [22], miRNAs [23], and SIRT [24] as potential signaling mechanisms. Recent work has demonstrated that TNFα induced senescent NP cells secrete soluble factors that are capable of inducing senescence in healthy NP cells [22], however the effect of these soluble factors on AF cells have not been assessed [25]. A further understanding of the impact of the NP senescent secretome on other disc cells is important to enable understanding of how the propagation of the degenerative phenotype within the disc occurs.

Reactive oxygen species (ROS), i.e., superoxide and $H_2O_2$, have also been shown to be capable of inducing senescence in a number of different cell types, including NP cells [10, 26] and AF cells [27]. $H_2O_2$ is a redox signaling factor and is produced by normal metabolizing cells. The $H_2O_2$ concentration in the cell is in the nanomolar concentration and outside of the cell ranges from approximately 1–5 μM [28, 29]. It signals, in part, by reversible oxidation of specific protein Cys thiolate residues which activates redox signalling. These can then activate phosphorylation cascades and transcription, to name a few processes [30]. $H_2O_2$ is produced through multiple pathways, one of which is the NADPH oxidase (NOX) family and the complexes of the electron transport chain [31]. NOX2 and 4 expression has been shown to increase within the IVD during degeneration in rats [32, 33], and in NP cells following exposure to IL1β or ROS in-vitro [32, 34]. Mitochondrial dysfunction has also been associated with IVD degeneration and has been proposed to contribute to ROS-induced damage within the tissues of the IVD [35]. In response to oxidative stress, mammalian cells have four enzymes that compose the primary ROS response: superoxide dismutase (SOD), catalase (CAT), glutathione peroxidase (GPX) [36], and peroxiredoxins (PRX). SOD catalyzes the dismutation of superoxide radicals to $H_2O_2$ which can then be degraded into $H_2O$ via CAT, GPX, or PRX. Cells in the intervertebral disc have been reported to express all isoforms of SOD [37, 38], although studies have consistently found a higher expression of SOD in the AF when compared to the NP [38, 39]. Decreased SOD and catalase activity/expression has been associated with IVD pathologies [40], which suggests that a redox imbalance within the IVD may contribute to the pathogenesis

of IVD degeneration [41]. Despite this, characterization of ROS scavenging systems within the disc remains poorly understood and has not been determined in the iAF.

Thus, the objective of this study is to identify factor(s) that induces iAF cell senescence. These studies will provide insight into factor(s) regulating iAF senescence and the contribution of inter-tissue communication on this process.

## Methods

### Cell isolation and monolayer culture

Intervertebral discs were aseptically excised from bovine caudal spines. IAF and NP tissues were visually distinguished and harvested as previously described [42–44]. NP and iAF tissues were each finely diced in HAM's F12 (318-010-CL, Wisent Bioproducts, Saint-Jean Baptiste, QC, Canada) into approximately 5 $mm^3$ cubes. Tissues were individually digested in 0.3% protease (Type XIV, Sigma-Aldrich, St. Louis, MO, USA) in HAM's F12 supplemented with 100 U/mL of penicillin-streptomycin (15140122, Gibco, Waltham, MA, USA) at 37°C for 1 hour, followed by 0.2% collagenase A (10103586001, COLLA-RO, Sigma-Aldrich, St Louis, MO, USA) in HAM's F12 supplemented with 100 U/mL of penicillin-streptomycin (15140122, Gibco, Waltham, MA, USA) at 37°C for 16 ±1 hours. Digested tissues were passed through 100 μm cell strainer and centrifuged at 800g for 8 minutes. Cells were washed three times in DMEM (319-016-CL, Wisent Bioproducts, Saint-Jean Baptiste, QC, Canada) supplemented with 5% fetal bovine serum (FBS; 080–150, Wisent Bioproducts, Saint-Jean Baptiste, QC, Canada). All cell counting was done manually, using a hemocytometer (HSR3110, Fisher Scientific, Waltham, MA, USA) and Trypan Blue (15250061, Gibco, Waltham, MA, USA). 17,000 cells/$cm^2$ of primary (P0) NP or iAF cells were plated separately in monolayer culture in T175 flasks (83.3912.502, Sarstedt, Rommelsdorf, Nümbrecht,, Germany) in DMEM supplemented with 5% FBS. P0 iAF and NP cells were cultured for 6 days and then passaged. Passage 1 cells were used in all experiments, unless otherwise stated.

In selected experiments, cells were grown overnight in serum-reduced DMEM (1% FBS) then treated with TNFα (40 ng/mL, 2279-BT, R&D Systems, Minneapolis, MN, USA, reconstituted in 0.1% bovine serum albumin in PBS) in DMEM supplemented with 5% FBS for 24 hours, followed by 5 washes with DMEM and then cultured cytokine-free in DMEM containing 5% FBS and allowed 24 hours to recover, prior to analysis. This concentration was chosen as it was shown previously to induce senescence in nucleus pulposus cells [21, 22].

### Non-contact co-culture

IAF and NP cells were isolated and cultured as described above. Monolayer P1 iAF and NP cells were used for all non-contact co-culture experiments. 14,000 cells/$cm^2$ iAF cells were seeded onto 12-well plates (83.3921.005, Sarstedt, Rommelsdorf, Nümbrecht,, Germany) and in separate cultures 14,000 cells/$cm^2$ NP cells were seeded onto hanging inserts (Costar 0.2 μm pore PTFE transwells, 3460, Corning, New York, NY, USA). This cell density was used so that cells were not confluent at the time of analysis. NP cells were treated with 40 ng/mL of TNFα for 24 hours, followed by 5 washes with DMEM only. Transwell inserts were then placed in the wells containing iAF cells and co-cultured for 24 hours in DMEM containing 5% FBS prior to analysis.

### Contact co-culture

Contact co-culture was performed as previously described [22] with some modifications. DMEM containing 5% FBS was used for all contact co-cultures, unless otherwise stated. NP

cells were passaged to P1, plated at 14,000 cells/cm$^2$ and treated with TNFα (40 ng/mL) for 24 hours. The iAF cells (P0) were cultured for 5–7 days until harvested using trypsin-EDTA for experimental set up. NP cells (P1) were trypsinized and resuspended in 20 μM CellTracker Red CMTPX dye (C34552, ThermoFisher, Waltham, MA, USA) in DMEM (319-016-CL, Wisent Bioproducts, Saint-Jean Baptiste, QC, Canada) according to the manufacturer's instructions. Greater than 95% labelling of cells was confirmed by fluorescent microscopy. NP and iAF cells were then mixed at a ratio of 1:1 and plated into chamber slides at 14,000 cells/ cm$^2$ (7,000 cells/cm$^2$ of each iAF and NP, or 14,000 cells/cm$^2$ of iAF alone) (81816, Ibidi, Grä-felfing, Bavaria, Germany). Experimental conditions were as follows: iAF cells alone, untreated NP and iAF coculture, and TNFα treated-NP and iAF coculture. In the final cocultures, NP cells were P2, and iAF cells were P1. Cells were cultured for 24 hours prior to analysis. Cells were evaluated for senescence by p16 immunocytochemistry as described below (06695248001, Roche CINtec, Basel, Switzerland). Cells were imaged using a Leica DMI-6000 spinning disc confocal microscope running Velocity imaging software. All contact co-culture quantification was assessed manually. P16 immunostaining was visualized in the far-red chan-nel, NP cells were labelled with CellTracker Red/DAPI, and iAF cells were positive for DAPI but were negative for CellTracker Red. Each image was assessed for number of p16$^+$ iAF, p16$^+$ NP, total iAF, and total NP cells. A minimum of 100 cells were assessed for each biological rep-licate. Results were displayed as the percentage of p16 positive iAF and NP cells in each condition.

## Formation of 3D tissue sheets

To form tissue, P1 iAF cells were seeded at high density (570k cells/cm$^2$) in monolayer culture (12-well plates, 83.3921.005, Sarstedt, Rommelsdorf, Nümbrecht, Germany). Tissues were cul-tured in DMEM supplemented with L-Proline (40 μg/mL, P0380, Sigma-Aldrich, St. Louis, MO, USA), Insulin-Transferrin-Selenium (1%, 315–080, Wisent Bioproducts, Saint-Jean Bap-tiste, QC, Canada), sodium pyruvate (1 mM, 600-110-EL, Wisent Bioproducts, Saint-Jean Bap-tiste, QC, Canada), and 10% FBS (complete medium). Non-adherent cells were removed after 2 days and replaced with complete medium with ascorbic acid (100 μg/mL, A4403, Sigma-Aldrich, St. Louis, MO, USA). The media was replaced with fresh complete media containing ascorbic acid every other day. Under these culture conditions, the cells accumulate extracellu-lar matrix molecules they synthesized and form tissue. These tissues were harvested after 10 days of culture. In selected experiments, on day 9 of culture the iAF tissue sheets were treated for 24 hours with TNFα (40 ng/mL) or 50 μM H$_2$O$_2$, followed by 5 washes with DMEM. The cultures were then placed in complete media without ascorbic acid and harvested 24 hours later.

## Histology and immunofluorescence

Monolayer cells were fixed in 4% paraformaldehyde (PFA) for 10 minutes and tissue sheets were fixed in 10% formalin for 12 minutes for histology and immunofluorescence and placed in 30% sucrose. Using a dissection microscope, agarose covered cell sheets were cut into thirds (each 12 mm wide). The center third of the tissue was mounted in OCT and cut cross-section-ally at 7 μm using a cryostat. Sections were collected onto silane coated slides and dried over-night at 40°C.

Sections were stained with hematoxylin and eosin, or Toluidine blue and cover-slipped using Micromount (3801731, Leica Biosystems, Nußloch, Baden-Württemberg, Germany). Tissues were imaged with a light microscope (Olympus BX61, Shinjuku, Tokyo, Japan) run-ning CellSens version 1.18.

For the tissue sheets, collagen type 1 and 2 immunohistochemistry was preceded by antigen retrieval using enzymatic digestion. Sections were incubated in Tris-buffered saline (TBS, pH 2) for 5 minutes, followed by pepsin (2.5 mg/mL in TBS pH2, P7012, Millipore Sigma, St. Louis, MO, USA) for 10 minutes at room temperature, followed by 3 washes with PBS. For aggrecan immunostaining the sections were incubated in hyaluronidase (25 mg/mL in PBS, pH 7.3, H3506, Millipore Sigma, St. Louis, MO, USA) for 30 minutes at 37˚C. Boiling sections in Dako Target Retrieval solution, (pH 9.0 S236784-2, Agilent Technologies, Santa Clara, CA, USA) for 10 minutes was used for MMP13 and p16 antigen retrieval. Monolayer cell staining did not utilize any antigen retrieval.

Sections or cells were then washed three times with PBS and blocked in 20% goat serum (16210–064, Gibco, Waltham, Waltham, MA, USA) and 0.1% Triton X-100 in PBS. Sections were incubated with primary antibody (listed in S1 Methods) at 4˚C overnight in a humidified chamber. The sections were washed three times with PBS before incubating with AlexaFluor secondary antibody (listed in S1 Methods) and 4'6-diamidino-2-phenylindole (DAPI) together at room temperature for 1 hour. Sections were washed 5 times with PBS and mounted with PermaFluor™ (TA030FM, ThermoFisher, Waltham, MA, USA). Negative controls consisted of replacing the primary antibody with a species matched IgG antibody at the same protein concentration (w/v). Immunofluorescence was imaged with a fluorescent microscope (IX81, Olympus, Shinjuku, Tokyo, Japan) and Velocity version 6.3.0. Quantification of ECM proteins in monolayer were assessed using ImageJ version 1.53q. COL1, COL2, p16, and ACAN were all stained independently and viewed in the red channel. The images were captured from the same 3 locations in each well using a well-overlay method (S1 Methods). Nuclear counting was automated by converting the image to binary and watershed separation. Positive cells (red cytoplasm) were counted manually.

Quantification of tissue thickness was calculated by measuring the average distance between the upper and lower edges at 4 standardized sites in the cross-section of the tissue using ImageJ version 1.53q. 2 sections approximately 0.2 mm apart and 2 images per section were assessed for all tissue analysis. 3 biological and 2 technical replicates were used for all tissue sheet quantification assays.

## Growth curves

To assess monolayer cell proliferation cells were grown for up to 72 hours in the presence or absence of TNFα, iAF cells were seeded in chamber slides at 14,000 cells/cm$^2$ (81816, Ibidi, Gräfelfing, Bavaria, Germany). Following 24 hours of no treatment or treatment with TNFα or serum starvation (as a positive control), iAF cells were fixed at 24 hour intervals over 3 days with 4% PFA. Cells were then stained for Ki 67 and counterstained with DAPI for 15 minutes in PBS. All cells were imaged in each well with a fluorescent microscope (IX81, Olympus, Shinjuku, Tokyo, Japan) and Velocity version 6.3.0. Nuclear counting was automated using ImageJ by converting the image to binary and watershed separation. All the cells were counted within the wells of each replicate. The cells did not undergo a media change during the 3 days the cell growth was evaluated.

## Senescence associated β-galactosidase staining

Senescence associated β-galactosidase activity (SA-βGal) was evaluated using the SA-βGal staining kit (#9860, Cell Signaling Technology, Danvers, MA, USA) according to the manufacturer's directions. Briefly, cells or tissues were fixed using solution composed of 2% formaldehyde/0.2% glutaraldehyde in PBS for 15 minutes. Cells were washed 3 times in PBS and incubated with the staining solution adjusted to pH 6 (40 mM citric acid/phosphate buffer,

5mM $K_4[Fe(CN)_6]$ $3H_2O$, 5mM $K_3[Fe(CN)_6]$, 150 mM sodium chloride, 2 mM magnesium chloride and 1 mg ml$^{-1}$ X-gal) in distilled water for 16 hours at 37˚C in a non-humidified oven. Cells were washed once with PBS, mounted with 70% glycerol and imaged under phase contrast microscopy (BX61, Olympus, Shinjuku, Tokyo, Japan, running CellSens version 1.18. 3). The images were captured from the same location in each well using a standardized well-overlay method (described in S1 Methods). Strong blue stained (SA-β-galactosidase-positive) cells were considered positive. ImageJ was used to count cells and the percentage of SA-β-galactosidase-positive cells (blue stained) was calculated.

## Quantification of secreted $H_2O_2$ by AmplexRed

Cells were seeded in 96-well plates at 14,000 cells/cm$^2$. Cells were cultured for 24 hours. In select experiments, cells were pretreated with the NOX inhibitor, 5 μM diphenyleneiodonium chloride (DPI) (D2926, Millipore-Sigma, St. Louis, MO, USA, resuspended to 5 mM in DMSO) in Hanks Balanced Salt Solution (HBSS) for 3 hours. Cells were subsequently treated with 50 μM $H_2O_2$ or 40 ng/mL TNFα for 16 hours. Cells were washed 3 times with DMEM and cultured treatment-free for 24 hours in DMEM containing 5% FBS. The media was removed and 105 μL of PBS was placed on the cells and placed back in the incubator for 1 hour (37˚C; 5% $CO_2$). The PBS supernatant was collected and $H_2O_2$ was quantified using AmplexRed assay (A22188, ThermoFisher, Waltham, MA, USA) according to the manufacturer's instructions. An $H_2O_2$ standard curve was created (0.0156 to 2 μM in PBS). Solutions were incubated with AmplexRed solution for 30 minutes in a black, flat bottom, 96-well plate (290-895-Z1F, Caplugs/Evergreen, Buffalo, NY, USA) at room temperature in the dark, and fluorescence intensity measured using EnSpire 2300 Multilabel Reader (running EnSpire Manager version 2.00) at 560/590 nm.

## CellROX green and JC-1 staining

To assess intracellular ROS and mitochondrial membrane potential CellROX green (C10444, ThermoFisher, Waltham, MA, USA) and JC-1 (T3168, ThermoFisher, Waltham, MA, USA) molecular probes were used, respectively. P1 NP and iAF cells were plated at 17,000 cells/cm$^2$ (6k cells/well) in 18-well Ibidi chambers (81816, Ibidi, Gräfelfing, Bavaria, Germany). Cells were treated with TNFα (40 ng/mL) or $H_2O_2$ (50 μM) for 16 hours, then washed three times with DMEM and cultured for 24 hours in DMEM supplemented with 5% FBS. Cells were then incubated with either CellROX green (5 μM) or JC-1 (5 μg/mL) according to the manufacturer's directions for 1 hour and visualized by epifluorescent microscopy (BX61, Olympus, Shinjuku, Tokyo, Japan). Quantification of CellROX green and JC-1 was done using ImageJ. CellROX green was analyzed for total fluorescence divided by the total number of cells. JC-1 was analyzed for average red fluorescence divided by average green fluorescence.

## Live/Dead and TUNEL staining

To assess viability of iAF or NP cells following exposure to TNFα, $H_2O_2$, and DPI, P1 NP and iAF cells were plated at 14,000 cells/cm$^2$ (6k cells/well) in 18-well Ibidi chambers (81816, Ibidi, Gräfelfing, Bavaria, Germany). Cells were pre-treated with DPI (5μM) or M40403 (100 μM) (10 mM in ethanol, Cayman Chemicals, Ann Arbor, MI, USA) resuspended in HBSS for 1 hour, washed 3 times with DMEM, followed by TNFα (40 ng/mL) or $H_2O_2$ (50 μM) in DMEM for 16 hours. In those experiments using DPI or M40403, control cells were pre-treated with the same amount of carrier (HBSS).

For the Live/Dead assay (L3224, Invitrogen, Waltham, MA, USA), cells were washed in DMEM and treated with Calcein-AM (2 μM) and Ethidium homodimer-1 (2 μM) diluted in

DMEM for 30 minutes and visualized by epifluorescent microscopy (BX61, Olympus, Shinjuku, Tokyo, Japan). 50 mM $H_2O_2$ in DMEM was used as a positive control.

For the TUNEL apoptosis assay (Terminal deoxynucleotidyl transferase (TdT) dUTP Nick-End Labeling (TUNEL) assay; 1168479591, Roche, Basel, Switzerland), cells in monolayer culture were stained according to the manufacturer's instructions. Cells were washed with PBS once and fixed with 2% PFA for 1 hour. Cells were rinsed once with PBS and permeabilized with the provided permeabilization solution for 2 minutes on ice. Label solution and enzyme solution were combined and incubated with the cells for 1 hour at 37°C. Cells were rinsed three times with PBS and mounted with PermaFluor™. Cells were incubated with the nuclear stain DRAQ5 (10 μM diluted in PBS, 62251, ThermoFisher, Waltham, MA, USA) for 15 minutes and visualized by epifluorescent microscopy (BX61, Olympus, Shinjuku, Tokyo, Japan).

### Gene expression

Cells in monolayer were placed in TRIzol (15596026, ThermoFisher, Waltham, MA, USA) and RNA isolated according to the manufacturer's instructions. For the tissue cultures, tissues were rinsed once with PBS and collected into TRIzol (1 mL, 12 well plate), vortexed briefly, and incubated for 10 minutes at which point the tissue was completely dissolved. RNA was isolated according to manufacturer's instructions. The pellet was washed with 75% ethanol overnight at -20°C. The following day, samples were spun at 7,500g for 5 minutes at 4°C. A second wash was performed with 75% ethanol and samples were air dried for 15 minutes and resuspended in 20 μL nuclease free water. RNA quantity and quality was assessed by spectrophotometer. Reverse transcription was performed using SuperScript III reverse transcriptase (18080093, ThermoFisher, Waltham, MA, USA) and 2 μg of RNA, according to the manufacturer's instructions. qPCR was performed using a Roche LightCycler 96. Primers are listed in S1 Methods. Gene expression analysis was presented as $2^{-\Delta Ct}$. To calculate ΔCt, technical replicates were averaged, and average 18S rRNA Ct values were subtracted from average Ct values of the gene of interest from the same biological replicate.

### Statistics

At least 3 biological replicates were obtained for each experiment, and 3 technical replicates/condition were performed unless otherwise specified. One biological replicate was composed of tissue from 3 intervertebral discs from a single bovine caudal spine. Unpaired T-test was used when comparing between two groups, and one-way or two-way analysis of variance (ANOVA) was used when comparing multiple conditions. To minimize family-wise type I error, Tukey's HSD post-hoc test was used when comparing multiple means. Significance was defined as $p < 0.05$. Analysis was done using GraphPad Prism Version 9.2.0.

### Ethics statement

As the disc tissue was obtained from the abattoir after euthanasia and was considered waste, no REB was required.

## Results

### TNFα induces an altered phenotype but not senescence in iAF cells

TNFα treated iAF cells had significantly more senescence associated β-galactosidase (SA-βGal) positive cells as compared to control (Fig 1A). However, it did not induce senescence as they retained their ability to proliferate as indicated by quantifying cell number and counting the

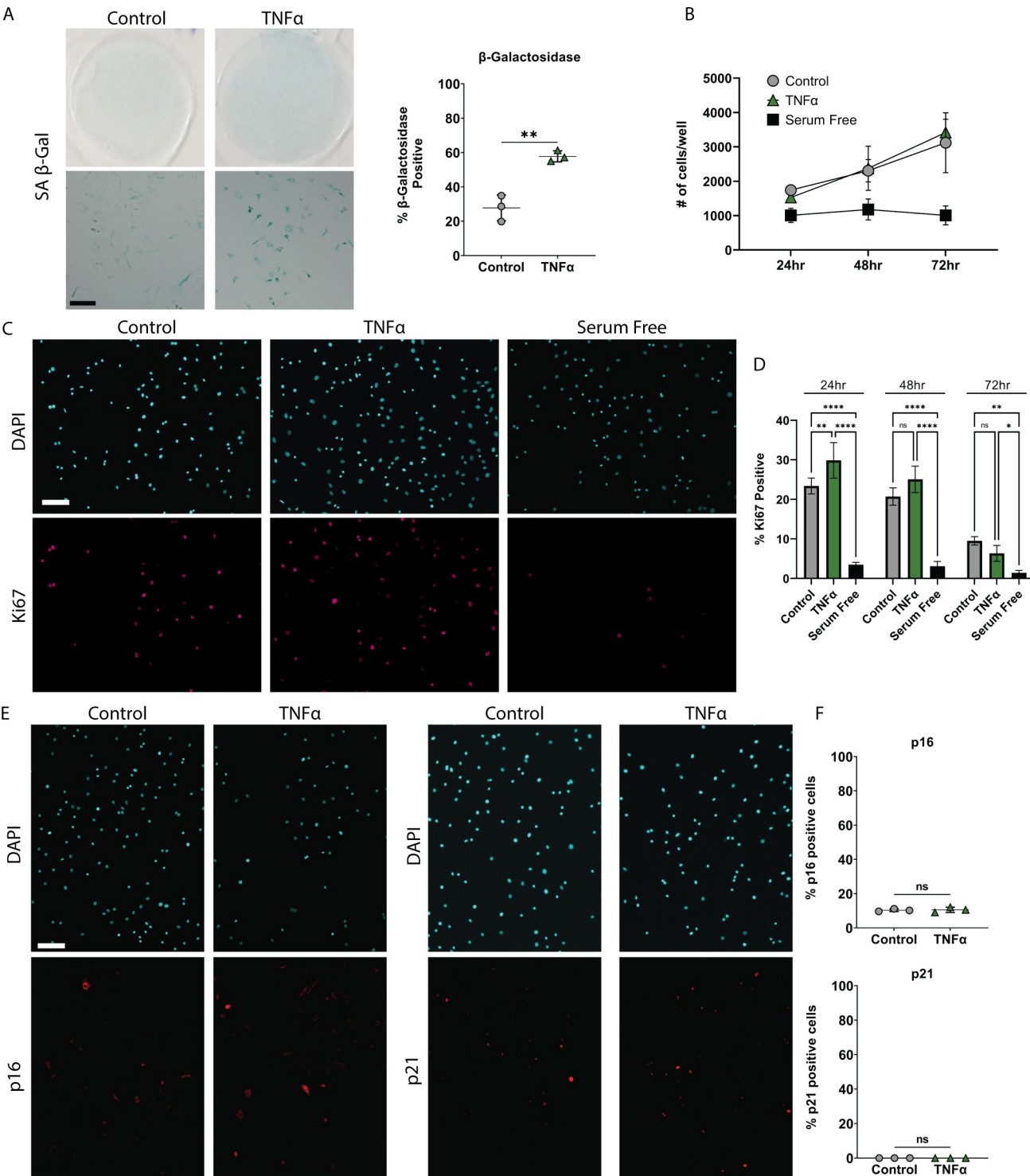

**Fig 1. iAF cells are resistant to TNFα induced senescence. (A)** Representative low and high magnification phase contrast images and quantification of senescence associated β-galactosidase staining of iAF cells cultured in the absence (control) or presence of TNFα. **(B)** Growth curve of iAF cells grown in the presence or absence of TNFα or serum starvation (control) at 24, 48, and 72 hours. **(C)** Representative images of Ki67 immunocytochemistry at the 24 hour time-point. **(D)** Quantification of percentage of Ki67 positive cells at 24, 48, and 72 hours post treatment. Media was not changed throughout the 72 hour time course. **(E)** Representative images of p16 and p21 immunostaining of iAF cells treated with TNFα. **(F)** Quantification of p16 and p21 immunostaining of iAF cells grown in the presence and absence (control) of TNFα, represented as percentage of total cells that were stained. Scale bars = 100µm. p <0.05 = *, p<0.01 = **, p<0.001 = ***, p<0.0001 = ****, N = 3 for immunostaining. Positive staining control images are shown in S4 Fig.

number of Ki67 positive immunostained iAF cells over a 3-day period (Fig 1B–1D, S1 Fig). TNFα exposure did not increase p21 or p16 accumulation in these cells, as determined by immunostaining (Fig 1E and 1F).

## iAF and NP cells have a differential ROS response following exposure to TNFα

As ROS have been implicated in mediating senescence and TNFα can lead to increased ROS production [9, 10], the effect of TNFα on iAF was examined. TNFα did not induce a change in ROS response in iAF cells. There was no increase in intracellular ROS as demonstrated by Cell-ROX green staining (Fig 2A and 2B) nor $H_2O_2$ secretion as quantified by AmplexRed assay (Fig 2C). Unlike iAF cells, NP cells exposed to TNFα, as a positive control, showed a significant increase in intracellular ROS accumulation and $H_2O_2$ secretion (Fig 2D–2F).

iAF cells had significantly higher expression of *SOD1* and *SOD2* following TNFα treatment when compared to their respective untreated cells. *CAT* expression was higher in untreated iAF cells than in TNFα treated iAF cells. In contrast, NP cells had no significant change in Superoxide dismutase *(SOD)-1*, *SOD2*, or catalase (*CAT)* gene expression upon exposure to TNFα. NADPH oxidase (*NOX) 1–5* gene expression was also assessed, however, only *NOX2* and *NOX4* were detectable in both the NP and iAF cells. *NOX2* but not *NOX4* expression was higher in TNFα treated iAF cells as compared to untreated iAF cells. Both *NOX2* and *NOX4* expression was significantly increased in TNFα treated NP cells when compared to untreated NP cells (Fig 2G and 2H).

Treating NP cells with diphenyleneiodonium chloride (DPI), a broad NOX inhibitor, caused a significant decrease in TNFα-mediated $H_2O_2$ accumulation (Fig 2I). iAF cells treated with DPI had no significant change in $H_2O_2$ accumulation (Fig 2J). DPI, at the concentration used, did not increase iAF or NP cell death as visualized by Live-Dead assay (S2 Fig).

## iAF cells express TNF-α receptors (TNFR1)

To determine if iAF cells express TNFR1, the gene and protein expression were evaluated. RT-PCR and western blot analysis show that iAF cells express TNFRI at both the gene and protein levels. Notably TNFRI gene expression was similar in both NP and iAF cells, however TNFR1 protein expression was higher in iAF compared to NP cells (Fig 3).

## iAF cells undergo senescence when exposed to low dose $H_2O_2$

To determine if iAF cells undergo senescence in response to $H_2O_2$, which could explain the lack of response to TNFα, iAF cells were exposed to low dose $H_2O_2$. $H_2O_2$ treated iAF cells in monolayer show increased staining for SA-βGal (Fig 4A), as well as an increased number of p16 and p21 immunopositive compared to control cells (Fig 4B and 4C). $H_2O_2$ did not induce cell death, as demonstrated by Live/Dead staining (Ethidium Homodimer/Calcein-AM), or apoptosis by TUNEL staining (S2 Fig). $H_2O_2$ exposure did not induce changes in *SOD1*, *SOD2*, *CAT*, *NOX2*, or *NOX4* gene expression (Fig 4D). Despite this, iAF cells exposed to $H_2O_2$ had a disrupted redox regulation with an increase in intracellular ROS and secreted $H_2O_2$, evaluated by CellROX green and AmplexRed assay, respectively (Fig 4E and 4F). $H_2O_2$ treated iAF cells also had depolarized mitochondrial membrane potential compared to untreated and TNFα treated cells, as determined by JC-1 red/green fluorescence intensity (Fig 4G).

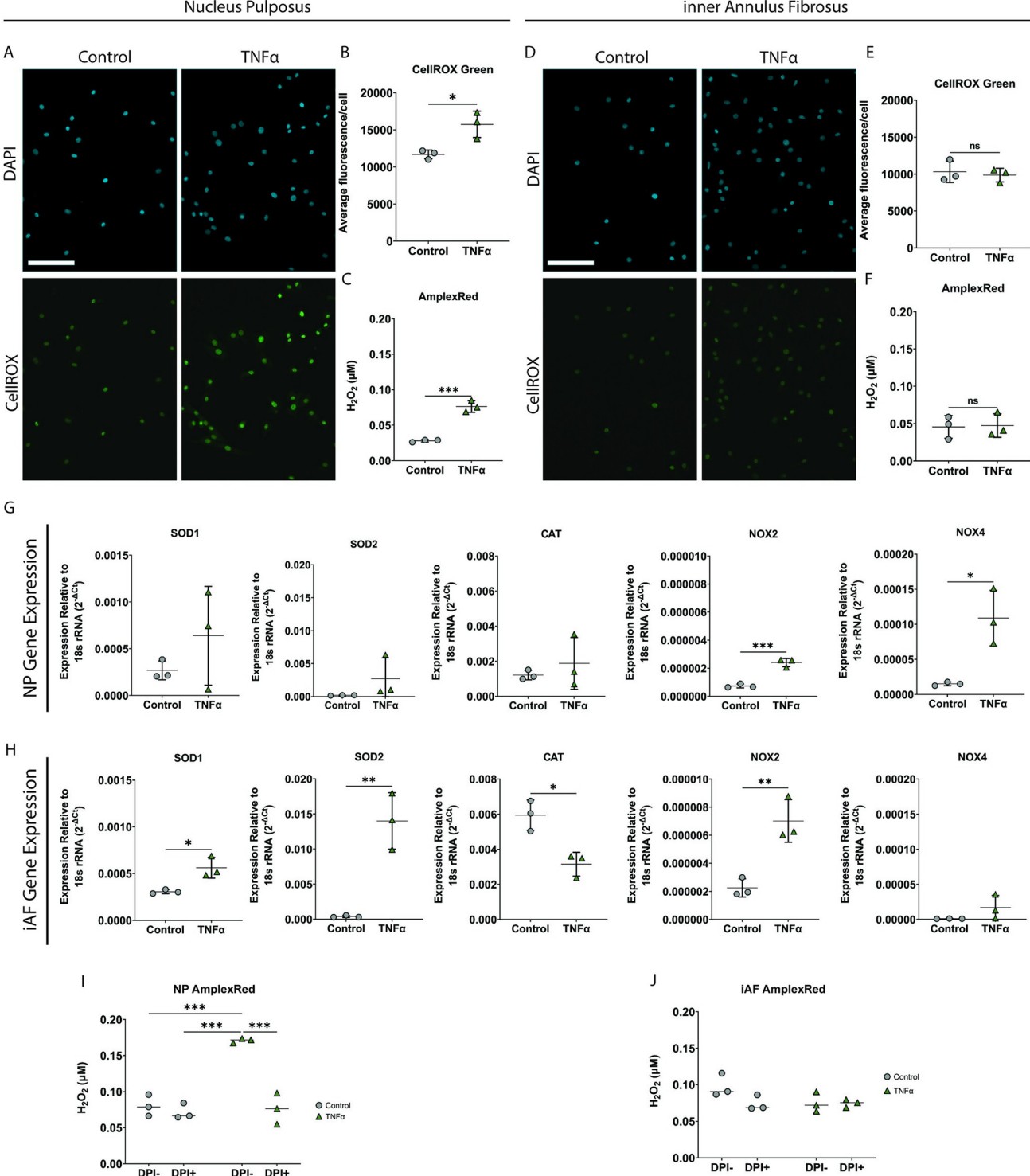

**Fig 2. iAF cells have a differential ROS response to TNFα compared with NP cells. (A)** Representative images of CellROX green staining in NP cells exposed to TNFα (40 ng/mL). **(B)** Quantification of CellROX green average fluorescence intensity per cell in NP cells exposed to TNFα. **(C)** AmplexRed assay quantification of $H_2O_2$ released from NP cells exposed to TNFα. **(D)** Representative images of CellROX green staining in iAF cells exposed to TNFα (40 ng/mL). **(E)** Quantification of CellROX green average fluorescence intensity per cell in iAF cells exposed to TNFα. **(F)** AmplexRed quantification of $H_2O_2$ released from iAF cells exposed to TNFα. **(G)** Gene expression analysis relative to 18s rRNA of ROS related genes in NP cells exposed to TNFα. 18S rRNA Ct value (mean±SD): Control = 9.56± 0.5, TNFα = 9.55±0.36. **(H)** Gene expression analysis relative to 18s rRNA of ROS related genes in iAF cells exposed to TNFα. 18S rRNA Ct value (mean±SD): Control = 8.93±0.80, TNFα = 9.86±0.36. **(I)** AmplexRed quantification of $H_2O_2$ released from NP cells

exposed to TNFα pre-treated with the NOX-inhibitor diphenyleneiodonium chloride (DPI). **(J)** AmplexRed quantification of $H_2O_2$ released from iAF cells exposed to TNFα pre-treated with the NOX-inhibitor DPI. Scale bars = 100μm. $p<0.05$ = *, $p<0.01$ = **, $p<0.001$ = ***, $p<0.0001$ = ****, N = 3 for all experiments.

## iAF cells treated with TNFα but not $H_2O_2$ undergo degenerative-like changes

IAF cells in monolayer treated with TNFα (40 ng/mL) showed a significant reduction in *COL2* and *ACAN* and an increase in *IL6* and *MMP13* gene expression (Fig 5A). This correlated with immunohistochemical data showing a significant reduction in the number of cells producing type II collagen and aggrecan when compared to untreated control cells (Fig 5B and 5C). Unlike TNFα, $H_2O_2$ (50 μM) exposure did not induce changes in *COL1*, *COL2*, *ACAN*, *IL6*, or *MMP13* gene expression (Fig 5D) or COL1, COL2, or ACAN immunopositivity (Fig 5E and 5F).

## iAF cells grown in 3D tissue sheets also undergo senescence in the presence of $H_2O_2$ but not TNFα

To determine if the inability of TNFα to induce senescence in iAF cells was an artefact of growing the cells in monolayer culture, cells were also grown in high density culture so they formed 3D tissue. These tissues contain collagen types I and II and aggrecan similar to native iAF (Fig 6A). As in monolayer, TNFα treated iAF tissue showed a significant increase in MMP13 protein as determined by immunostaining. The iAF tissue sheets had a significant increase in the number of p16 immunoreactive cells when exposed to $H_2O_2$ but not TNFα (Fig 6B). Although TNFα and $H_2O_2$ treated iAF tissues showed a significant decrease in thickness compared to untreated controls, the decrease was greater in the cytokine treated tissues (Fig 6C).

## iAF cells have an altered phenotype following exposure to media conditioned by TNFα treated-NP cells, but only undergo senescence in a contact co-culture model

To determine if iAF cells can respond to the TNFα treated-NP secretome, iAF cells were co-cultured with either TNFα or untreated-NP cells in a contact or non-contact co-culture system. iAF cells in non-contact co-culture with TNFα treated-NP cells have a significant increase in *IL6* and *MMP13* gene expression (Fig 7A) as well as a reduction in the number of type II

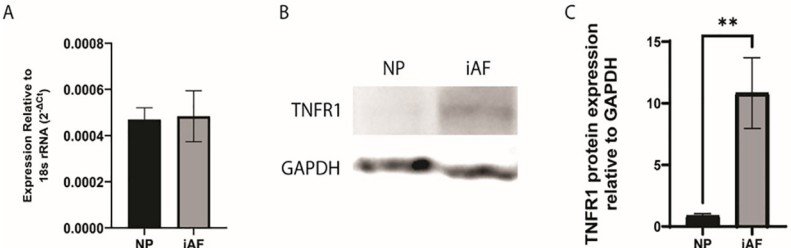

**Fig 3.** (A) Gene expression was determined by real time reverse-transcription-PCR (RT-PCR). The results were normalized to 18S rRNA (ΔCt) and expressed as mean ± SD (N = 3). (B and C) Expression of TNF receptor 1 (TNFR1) by nucleus pulposus (NP) and inner annulus (iAF) cells were evaluated by Western blot analysis (B) and quantified by densitometry (C). The expression was normalized by the loading control, GAPDH. (N = 3; ** $p<0.01$ relative to NP cells).

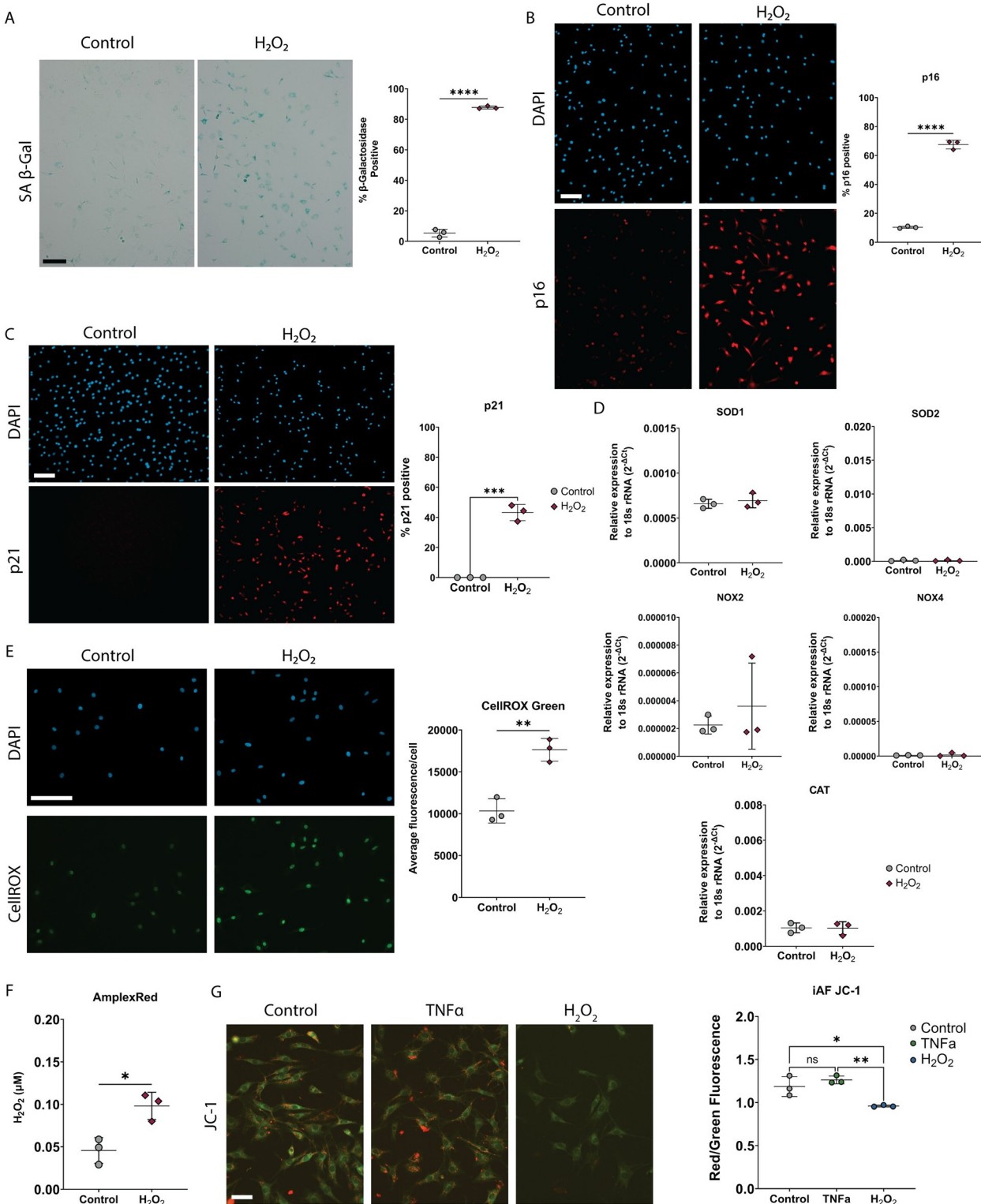

**Fig 4. iAF cells undergo H$_2$O$_2$-induced senescence.** **(A)** Representative phase contrast images and quantification of senescence associated β-galactosidase staining of iAF cells treated with H$_2$O$_2$ (50 μM). **(B)** Representative images and quantification of p16 immunostaining of iAF cells treated with H$_2$O$_2$. **(C)** Representative images and quantification of p21 immunostaining of iAF cells treated with H$_2$O$_2$. **(D)** Gene expression analysis of iAF cells treated with H$_2$O$_2$. 18S rRNA Ct value (mean±SD): Control = 7.76± 0.16, H$_2$O$_2$ = 7.75± 0.18. **(E)** Representative images and quantification of intracellular ROS with CellROX green. **(F)** AmplexRed quantification of H$_2$O$_2$ released from iAF cells exposed to H$_2$O$_2$. **(G)**

Representative images and quantification of JC-1 staining in iAF cells exposed to TNFα or $H_2O_2$. Scale bars = 100μm. p <0.05 = *, p<0.01 = **, p<0.001 = ***, p<0.0001 = ****, N = 3 for all experiments.

collagen immunopositive cells compared to co-culture with untreated-NP cells. (Fig 7B). There was no change in *COL1* or *ACAN* gene and protein expression. IAF cells co-cultured with TNFα-treated-NP also had a significant increase in SA-βGal positive cells (Fig 7C) but were negative for p21 and p16 immunoreactivity (Fig 7D).

In a contact co-culture system, iAF cells admixed with TNFα treated-NP cells, underwent senescence as there was significantly more p16 immunopositive cells when compared to iAF cells co-cultured with untreated NP cells (Fig 7E).

## Discussion

Although TNFα is known to induce senescence in NP cells, this study demonstrates that iAF cells are resistant to TNFα-induced senescence under the conditions examined even though the cells expressed TNFα receptor 1. Previous reports have suggested that increased ROS-accumulation is critical for TNFα-induced senescence [45]. iAF cells did not increase intracellular ROS or $H_2O_2$ secretion in response to TNFα. Additionally TNFα treated iAF cells had increased expression of the superoxide scavengers *SOD1* and *SOD2* which may explain why TNFα-did not induce senescence. Interestingly, TNFα treated NP cells had increased expression of *NOX4* which has been shown to produce $H_2O_2$ [46]. Given that iAF cells were resistant to TNFα induced senescence, potentially through maintenance of ROS homeostasis, we next investigated the effect of exogenous low dose $H_2O_2$ on iAF cells. $H_2O_2$ did induce senescence in iAF cells, however unlike TNFα, $H_2O_2$ did not induce the release of markers typically associated with matrix degeneration, as there was no change in *COL2*, *ACAN*, *MMP13*, or *IL6* gene expression nor the number of cells producing type II collagen and aggrecan. This suggests that iAF cells may have distinct degenerative and senescent phenotypes. Lastly, as studies have demonstrated that degenerative changes occur in the NP prior to the AF [47, 48] and that senescent NP cells secrete soluble factors capable of inducing senescence in healthy NP cells [22], we next investigated if the senescent NP secretome is capable of inducing senescence in iAF cells. Co-culturing iAF cells with TNFα-induced senescent NP cells, did induce senescence of the iAF cells, as well as inducing some degenerative changes when compared to co-cultures with untreated NP cells. Thus, senescent NP cells may contribute to the senescent and degenerative changes observed within the iAF during IVD degeneration.

Although $H_2O_2$ accumulation within the disc is well characterized [35, 49–51], to our knowledge this is the first study to demonstrate that $H_2O_2$, and not the cytokine TNFα, may be a principle driver of senescence in the iAF and importantly, that it may act as a signaling molecule between the NP and iAF. $H_2O_2$ can be transported across the plasma membrane through aquaporins, which are known to be expressed by both NP and AF cells [52, 53], and has been proposed to facilitate cell-to-cell signaling [54]. The crosstalk between NP and iAF cells is not entirely unexpected as a previous study has demonstrated $H_2O_2$ can be secreted by one cell type and taken up by another in-vitro [54]. Given that iAF cells are sensitive to low dose $H_2O_2$, and that TNFα treated NP cells increased the amount of secreted $H_2O_2$, this may suggest that the $H_2O_2$ generated from TNFα-induced senescent NP cells may be responsible for the senescence observed in the co-cultured iAF cells. Interestingly, iAF cells co-cultured with NP cells in a non-contact culture system did not undergo senescence. While there may be the need for direct contact to enable the senescent effects of $H_2O_2$ on iAF cells, it is possible that it just reflects the unstable nature of $H_2O_2$. Studies in the literature, using plant cells,

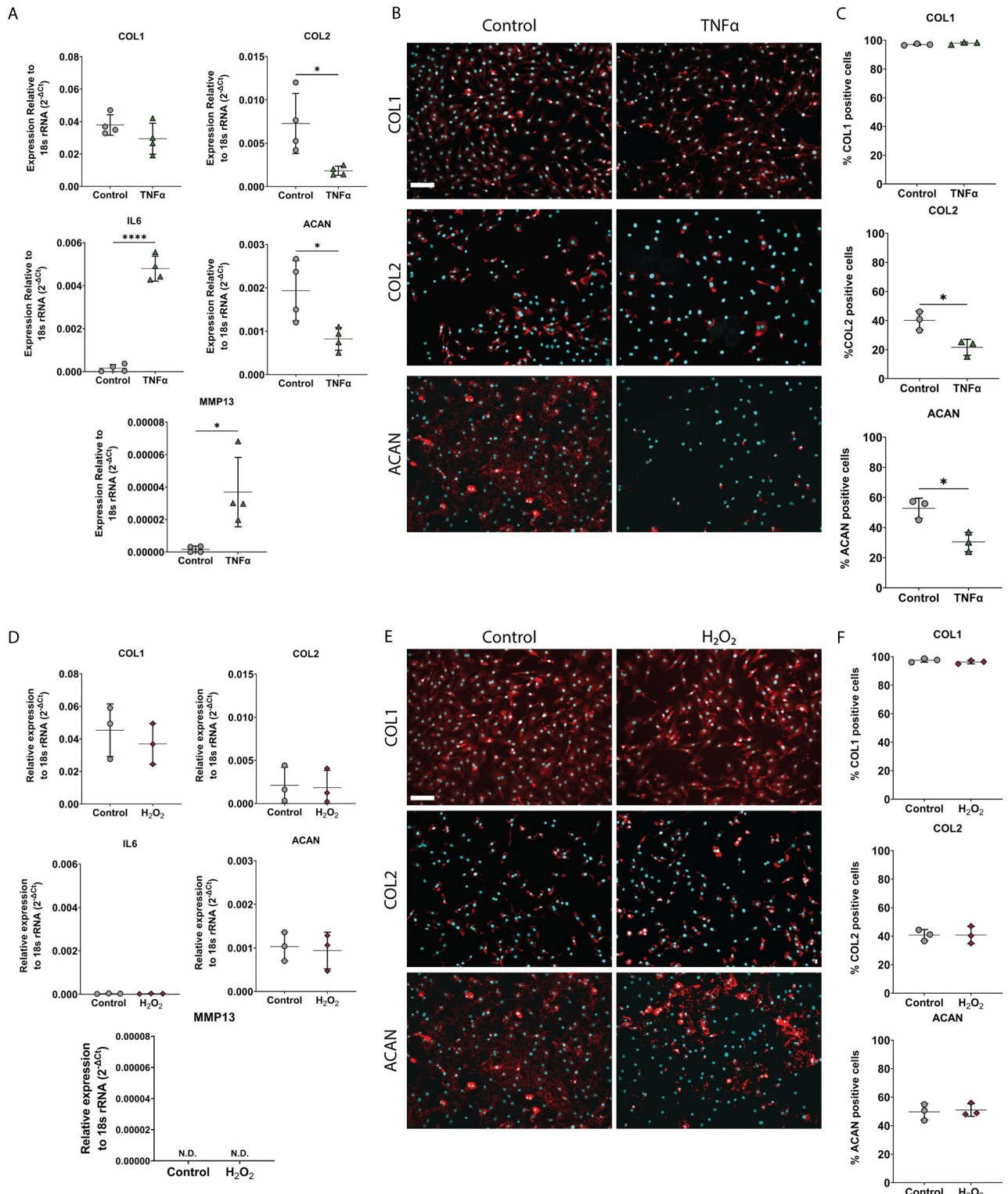

**Fig 5. TNFα treated but not H₂O₂-induced senescent iAF cells show signs of degeneration at 24hrs. (A)** Gene expression analysis of iAF cells treated with TNFα (40 ng/mL). 18S rRNA Ct value (mean±SD): Control = = 9.37± 0.28, TNFα = 9.72±0.21. **(B)** Representative images of COL1, COL2, and ACAN immunocytochemistry of iAF cells treated with TNFα. **(C)** Quantification of immunocytochemistry in B, presented as percentage of total cells stained. **(D)** Gene expression of iAF cells treated with H₂O₂. 18S rRNA Ct value (mean±SD): Control = 7.62± 0.16), H₂O₂ = 7.47± 0.28. **(E)** Representative images of COL1, COL2, and ACAN immunocytochemistry of iAF cells treated with H₂O₂. **(F)** Quantification of immunocytochemistry

in E, presented as percentage of total cells stained. Scale bars = 100μm. p <0.05 = *, p<0.01 = **, p<0.001 = ***, p<0.0001 = ****, N = 3 for all experiments. Positive staining controls are shown in S4 Fig.

demonstrated that $H_2O_2$ diffusion distance within a cell is just on the order of 1μm, with an approximate half life of just 1ms [55, 56]. If in the non-contact culture $H_2O_2$ levels were not high enough by the time it diffused to the iAF cells or present for sufficient time to induce senescence in the iAF cells under the conditions examined, the effects of $H_2O_2$ may not occur.

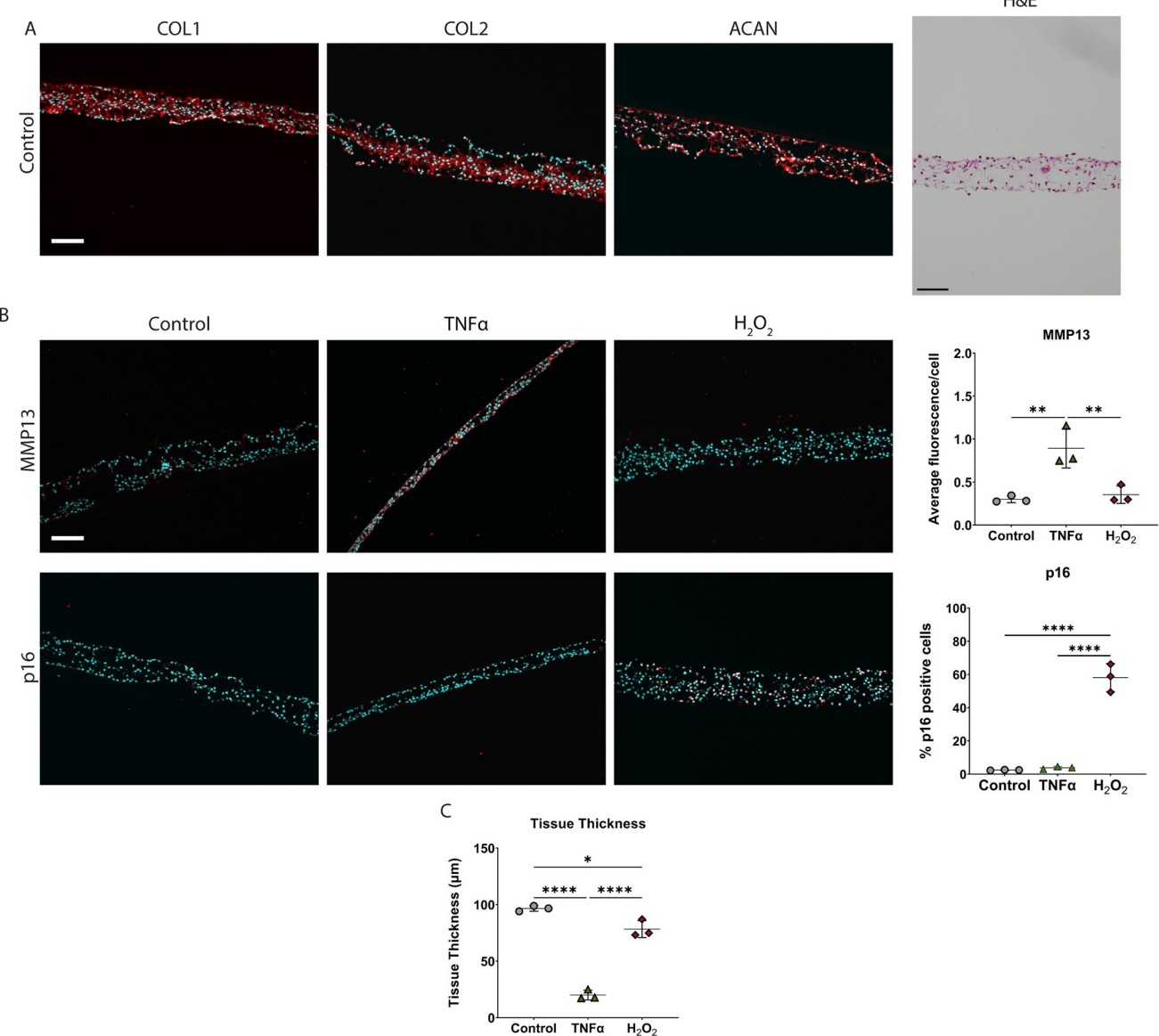

**Fig 6. iAF cells in 3D tissue sheets undergo senescence when exposed to $H_2O_2$ but not TNFα. (A)** Representative images of untreated iAF tissue sheets following immunohistochemical staining for COL1, COL2, or ACAN, or H&E staining. **(B)** Representative images of immunohistochemistry and quantification of MMP13 and p16 of iAF tissue sheets exposed to TNFα (40 ng/mL) or $H_2O_2$ (50 μM) for 24 hours. **(C)** Average cross-sectional thickness of iAF tissue sheets exposed to TNFα or $H_2O_2$. Scale bars = 100μm. p<0.05 = *, p<0.01 = **, p<0.001 = ***, p<0.0001 = ****, N = 3.

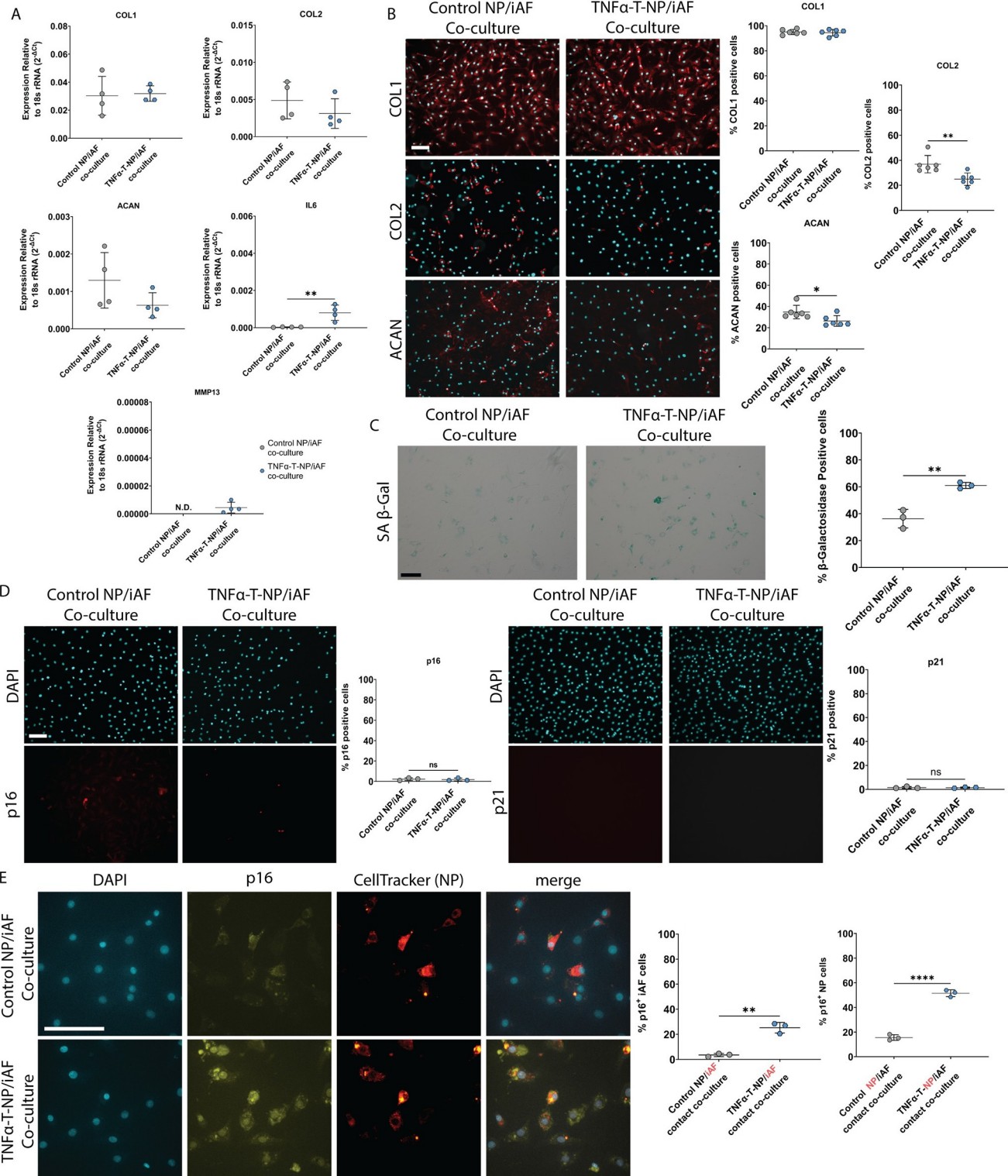

**Fig 7. iAF cells have an altered phenotype when co-cultured with senescent NP cells. (A)** Gene expression of iAF cells exposed to non-contact co-culture with TNFα or untreated NP cells for 24 hours. 18S rRNA Ct value (mean±SD): Control = 8.75±0.28, TNFα = 8.93±0.27. **(B)** Representative images and quantification of COL1, COL2, and ACAN immunostained iAF cells exposed to non-contact co-culture with NP cells. **(C)** Representative phase contrast images of senescence associated β-galactosidase staining and quantification of iAF cells exposed to non-contact co-culture with NP cells. **(D)** Representative images and quantification of p16 and p21 immunocytochemistry of iAF cells exposed to non-contact co-culture with NP cells. **(E)** Representative images

and quantification of iAF cells in a contact co-culture with TNFα and untreated NP cells. Scale bars = 100μm. p<0.05 = *, p<0.01 = **, p<0.001 = ***, p<0.0001 = ****, N = 6 for collagen and aggrecan immunostaining, N = 3 for all other immunostaining, N = 4 for gene expression. TNFα-T-NP = TNFα treated NP cells.

Alternatively, as $H_2O_2$, or even the ROS generating NADPH oxidase, can be transported via exosomes [57, 58], it is possible that diffusion of exosomes was impaired through the transwell pores. Nevertheless, taken together this data indicates that $H_2O_2$ transport within the disc may play a role in senescence propagation between cell types. This contrasts with the dominant theory that propagation of senescence from the NP to AF is due to NP degeneration that leads to aberrant ECM remodelling and subsequently compromise of mechanical properties [59]. While the latter is likely a contributing cause to the degradation of the tissues as altered biomechanical stimuli within the AF tissues have been shown capable of inducing senescence in AF cells [60], it may not be the initiating factor.

Previous studies have identified resistance to TNFα mediated cytotoxicity, however cellular senescence was not investigated in those studies. It is possible that the mechanism of resistance to TNFα cytotoxicity may be similar to the mechanism of senescence resistance in iAF cells. Resistance to TNFα in embryonic KYM-1 cells appears to be mediated by the loss of TNFα-receptor expression [61] or in HEK/HeLa cells by secretion of TNFα neutralizing proteins [62]. The former is an unlikely explanation as iAF cells were shown to express TNFR1 and are able to respond to TNFα treatment, as demonstrated by alterations in gene expression and accumulation of ECM components. Alternatively, others have found that differential phospholipase A2 activation altered the ability of TNFα to induce cytotoxicity in epithelial, ovarian, and cervical cell lines [63, 64]. Phospholipase A2 activation is thought to lead to the production of ROS that can be mitigated by superoxide dismutase (SOD). SOD-mediated protection has been demonstrated in L929 cells and L929.12 cells which are sensitive and resistant to TNFα cytotoxicity, respectively [65]. This was validated by studies in epithelial cells [66] where it was shown that SOD2 over-expression protects cells from TNFα-mediated ROS cytotoxicity. Although these studies have not investigated the senescence response of the TNFα-resistant cells, our study has similar findings with respect to the regulation of ROS, potentially via SOD.

It is not entirely unexpected that iAF cells respond differently to TNFα, than NP cells. The lack of senescence response to TNFα treatment is not due to the absence of the TNFR1 as iAF cells express the protein at levels higher than NP cells. However, the response of cells to TNFα following receptor binding may be pleiotropic given the complex signaling pathways that are activated after this interaction occurs. This can include sequential formation of TNFR1 complex 1 and the cytosolic TNFR1 complex II [67] which can be further modified in many different ways, for example by tyrosine kinases, IKK, or via crosstalk between other signaling pathways such as Hippo [68]. Furthermore, co-binding with other cytokines such as IL12, or TGFβ [69] can also influence the outcome. Secondly, although the NP and iAF cells share some phenotypic similarities, the NP and AF are derived from different embryonic lineages: the NP is from the notochord and the AF from the paraxial mesoderm [70]. Furthermore, in adult IVD tissues, these cells still have distinct transcriptomes that may alter their responsiveness to TNFα [39]. Previous work has identified *SOD2* as one of the top differentially expressed genes between iAF and NP cells, basally and in response to TNFα stimulation [38, 39]. This difference in *SOD2* is consistent with our findings, as iAF cells were shown to have higher TNFα-induced expression of *SOD1* and *SOD2* as compared to NP cells. The *SOD2* response to TNFα is not found across all cell types, since a reduction in *SOD2* expression in response to TNFα has been reported in mouse embryonic fibroblasts [71]. In the current study, evaluation of ROS accumulation in iAF and NP cells showed a significant increase in intracellular ROS in

NP but not iAF cells following TNFα-exposure, which is consistent with iAF cells having greater regulation of superoxide and $H_2O_2$ [72]. There are numerous proteins that regulate cellular ROS effects that were not specifically assessed in this study. Most notably glutathione-peroxidase, thioredoxin, peroxiredoxin, and glutathione pathways. To date, these molecules have largely been understudied in IVD cells. Specifically, two studies have indicated that AF and NP cells may decrease their expression of GPX and GSH when exposed to ROS stress or in menopause mediated IVD degeneration [73, 74]. Alternatively, other studies have identified non-coding RNAs such as NKILA/miR-21 [75, 76], and cellular proteins such as Sirt6 [77], or prolonged NF-κB activation can increase resistance to the cytotoxic effects of TNFα [78]. This suggests that other mechanisms of TNFα signaling regulation may play a role in the iAF cells resistance to TNFα mediated senescence. Further studies are required to elucidate the mechanism(s) regulating this lack of senescent response by iAF cells to TNFα treatment.

Despite increased senescence associated β-galactosidase positivity in iAF cells following exposure to TNFα or non-contact co-culture NP cells, these cells remained p16 and p21 negative–suggesting they are not senescent. Senescence associated β-galactosidase (SA-βGal) was the first stain reported in the literature to identify senescent cells [79], however on its own it is not sufficient to confirm senescence [10]. The assay is designed to measure the endogenous lysosomal β-galactosidase activity outside the enzyme's ideal pH range. By using this pH, only cells that express significant amounts of lysosomal β-Galactosidase are stained. Interestingly, although high SA-βGal activity has been associated with senescence it has been shown that lysosomal β-Galactosidase is not required for the senescence program [80]. Additionally, many studies have found SA-βGal staining in the absence of senescence, such as in high cell density culture and serum starvation [81]. Other factors have also been shown to induce a positive βGal stain in the absence of senescence such as tartrate-resistant acid phosphatase (TRAP) expression in osteoclasts [82, 83] or lysosomal activity in young neurons [84].

This study has some limitations. Specifically, as this is an in-vitro investigation, the manner by which these cell types interact in this environment may not accurately reflect in-vivo signaling. ROS are capable of reacting with ECM components and integrins [85], which may sequester $H_2O_2$ before it reaches adjacent cell types, although this only enhances the case made for the need for direct cell contact. Similarly, other ligands that may modulate TNFα or ROS responsiveness are known to be sequestered by ECM components, such as TGFβ [86, 87]. ECM components are also known to alter cellular responsiveness to ligands [88, 89] and ROS production [85]. Extracellular ROS scavenging enzymes may also inhibit ROS/$H_2O_2$ from acting as a signaling molecule within the disc, and these may decrease with increasing degeneration [40]. Secondly, the concentrations of TNFα (40ng/mL) and $H_2O_2$ (50μM) used in this study are significantly higher than concentrations found in-vivo (TNFα: 0.05–0.15ng/mL, $H_2O_2$: 0.25μM [18, 90]). Thirdly, sample size for most of this study was at least 3 independent biological samples (N = 3) repeated in triplicate. It is possible that this sample size may lead to type I and II errors. Given that some of the effect sizes, although significant, were limited, the biologic relevance of these findings will require further investigation. Finally, NP/iAF co-cultures were performed at a cell ratio of 1:1, which is higher than the NP:iAF ratio in vivo in bovine caudal intervertebral discs.

In summary, the results suggest that iAF cells are sensitive to the senescent effect of $H_2O_2$. The cells appear to be resistant to TNFα-induced senescence at the concentration evaluated, perhaps due to their inability to produce sufficiently increased $H_2O_2$ levels on exposure to the cytokine. Interestingly, NP cells exposed to TNFα undergo senescence and secrete factors that induce degenerative and senescent changes in iAF cells. Disc degeneration has been shown to be multifactorial, with cytokines, genetics, mechanics, and environmental stressors all contributing to the pathophysiology [91–93]. The current study reported here, provides another

potential contributing mechanism for the positive feedback loop of disc degeneration, specifically through ROS accumulation and the capacity for NP to signal iAF cells, an action which could promote degenerative and senescent changes. This could inform the choice of IVD degeneration therapeutics. For example, targeting only senescent iAF cells may not fully ameliorate degeneration of this tissue, but reducing NP cell capacity to produce ROS (e.g., NOX knockout), or inflammatory cytokines (e.g., TNFα knockout) to prevent the positive feedback of inflammatory and oxidative stress may be equally if not more efficacious. Further study is required to determine if human disc cells respond similarly.

## Supporting information

**S1 Fig. Representative images of Ki67 immunocytochemistry of iAF cells treated with TNFα (40 ng/mL) or serum starved at 48 and 72 hours of treatment.** DAPI was used as a nuclear stain. Scale bar = 100 μm.
(TIF)

**S2 Fig. iAF cell viability when exposed to TNFα, $H_2O_2$, or DPI. (A)** Representative images of TUNEL staining with DRAQ5 as a nuclear stain and ethidium homodimer (red)/Calcein-AM (green) following exposure to TNFα (40 ng/mL) or $H_2O_2$ (50 μM). Positive controls were exposed to DNAse I for TUNEL staining and 50 mM $H_2O_2$ for ethidium homodimer/calcein-AM staining. **(B)** Quantification of percentage of TUNEL positive iAF cells exposed to TNFα and $H_2O_2$. **(C)** Representative images of ethidium homodimer (red)/Calcein-AM (green) stained iAF and NP cells following exposure to diphenyleneiodonium chloride (DPI) (5 μM). Student's T-test was used for statistical analyses of DPI study, one-way ANOVA with Tukey's post-hoc was used for iAF TUNEL and iAF live-dead quantification. N = 3 for all experiments. Scale bars = 100 μm.
(TIF)

**S3 Fig. Senescence markers in iAF and NP cells isolated from the same disc and stained at the same time in the presence or absence of TNFα.** (A) Representative images and quantification of SA-βGal stained untreated controls and TNFα treated iAF and NP cells. (B) Representative images and quantification of p16 stained untreated controls and TNFα treated iAF and NP cells. $p < 0.05$ = *, $p < 0.01$ = **, $p < 0.001$ = ***, $p < 0.0001$ = ****, N = 3; Scale bar = 60 μm.
(TIF)

**S4 Fig. Immunohistochemical positive control images of (A) NP cells stained with antibody reactive with p16 following 24hr exposure to TNFα.** (B-C) Tissue formed by nucleus pulposus cells grown for 2 weeks in 3D culture stained for ACAN (B) and COL2 (C). (D) Native bovine IVD tissue stained with antibody reactive with COL1. DAPI was used as a nuclear counterstain. Scale bar = 100 μm.
(TIF)

**S1 Methods.**
(DOCX)

## Acknowledgments

We would like to thank Drs. Marco Massina and Christopher Chang for providing PG1-FM and insights on ROS signaling. We would also like to thank Ryszard Bielecki and Louise Brown for their microscopy expertise and Drs. Darcy Lidington and Aleksandra Radchenko for providing the TNFR1 primers and antibody.

## Author Contributions

**Conceptualization:** Aaryn Montgomery-Song, Sajjad Ashraf, Paul Santerre, Rita Kandel.

**Data curation:** Aaryn Montgomery-Song, Sajjad Ashraf.

**Formal analysis:** Aaryn Montgomery-Song, Sajjad Ashraf, Paul Santerre, Rita Kandel.

**Funding acquisition:** Paul Santerre, Rita Kandel.

**Investigation:** Aaryn Montgomery-Song, Sajjad Ashraf.

**Methodology:** Aaryn Montgomery-Song, Sajjad Ashraf.

**Project administration:** Paul Santerre, Rita Kandel.

**Resources:** Paul Santerre, Rita Kandel.

**Supervision:** Paul Santerre, Rita Kandel.

**Validation:** Aaryn Montgomery-Song, Sajjad Ashraf.

**Visualization:** Aaryn Montgomery-Song, Sajjad Ashraf, Paul Santerre, Rita Kandel.

**Writing – original draft:** Aaryn Montgomery-Song.

**Writing – review & editing:** Aaryn Montgomery-Song, Paul Santerre, Rita Kandel.

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
