## [Decision Letter · Decision Letter 0]

14 Feb 2023

PONE-D-22-34265Senescent response in inner annulus fibrosus cells in response to TNFα, H2O2, and the nucleus pulposus SASP secretomePLOS ONE

Dear Dr. Kandel,

Thank you for submitting your manuscript to PLOS ONE. After careful consideration, we feel that it has merit but does not fully meet PLOS ONE’s publication criteria as it currently stands. Therefore, we invite you to submit a revised version of the manuscript that addresses the points raised during the review process.

We look forward to receiving your revised manuscript.

Kind regards,

Svenja Illien-Jünger, Ph.D.

Academic Editor

PLOS ONE

Journal Requirements:

2. Please amend either the title on the online submission form (via Edit Submission) or the title in the manuscript so that they are identical.

4. Please include a caption for figures 5 and 6.

Reviewers' comments:

Reviewer's Responses to Questions

**Comments to the Author**

1. Is the manuscript technically sound, and do the data support the conclusions?

Reviewer #1: No

Reviewer #2: No

2. Has the statistical analysis been performed appropriately and rigorously? 

Reviewer #1: No

Reviewer #2: Yes

3. Have the authors made all data underlying the findings in their manuscript fully available?

Reviewer #1: Yes

Reviewer #2: Yes

4. Is the manuscript presented in an intelligible fashion and written in standard English?

Reviewer #1: Yes

Reviewer #2: Yes

5. Review Comments to the Author

Reviewer #1: The study “Senescent response in inner annulus fibrosus cells in response to TNFα, H2O2, and TNFα-induced nucleus pulposus senescent secretome” is evaluating an interesting and clinically relevant topic. It is proposing a novel difference between NO and iAF cells, but the data is not quite there to support the theory.

General comments,

- The paper is aiming to compare NP and iAF cells but both cell types are often not included in the same set of experiments. Please correct.

-Please change the title to reflect the study.

-The references are outdated and don’t reflect the current state of the art. Please update.

-A positive control for each AB should be included in each experiment for confirmation.

-In general, the IHC are of low quality and are not reflecting the quantifications. Please correct.

-The figure legends in the text are mislabeled. For example, there are 3 figure legends labelled Fig 1. Please correct.

-The supplementary figure S2 is not mentioned in the text.

-The number of biological replicates seems too low especially since there is significant variability between the 3 biological replicates.

Specific questions,

Figure 1. It is clear from the literature that SA-βGal is overestimating and p16 underestimates the number of senescent cells, but I have never seen a disconnect of this magnitude. It would help interpret the data if NP and iAF cells were treated and analyzed simultaneously.

Figure 1, B, please provide the same analysis for NP cells. The cells that escaped senescence would continue to proliferate and comparing the two cell types would indicate if the presence of senescent cells affects the continued proliferation.

E, F, there are no visible p16-positive cells, how can about 10% detected? Please provide images that reflect the quantification.

Figure 2. The y-axis should have the same scale across the analysis to allow for comparison.

B and E. The iAF data is much more variable than NP and would need more biological repetitions to give solid data. Its a detail but it seems that the image in 2D is not the same for Dapi and CellROX

G-H The data is too variable to make the proposed conclusions. PCR data usually requires a minimum of n=6 to give solid data. Was a power analysis performed? I would also suggest presenting the data normalized to its own untreated control (2–∆∆Ct). Please mention the ct values of the housekeeping gene under the different treatments. Please add more biological replicates.

Figure 3. Please show data for NP and iAF cells to facilitate comparison and validate the conclusion.

B-C please show images reflective e of the quantification.

Figure 4A, Please see comments in 2G-H

Figure 5. Please show data for NP and iAF cells to facilitate comparison and validate the conclusion.

Please adjust the discussion in light of the additional experiments.

Reviewer #2: The authors investigated the effect of H2O2 and TNFα on iAF and NP cells and analyzed the changes in the production of the extracellular matrix components as Col1, Col2, ACAN. Knowing differences between disc cell compartments and how the specific iAF and NP cell types are affected by TNFα and H2O2 as models of cell senescence is highly relevant. However, the results presented in the manuscript are not enough to clearly demonstrate the differential effects of H2O2 and TNF on NP and iAF. My suggestions including a detail timeline of the NP and iAF cultures, iAF and NP cell characterization at the expression level, curves of Dose- effect of H2O2 and TNF specific for iAF and NP, time effect to recognize the ability of the iAF and NP cells to respond to the stressors (TNFα, H2O2). Unfortunately, immunostaining and in general images doesn’t reflect the quantification results. I strongly suggest high magnification in the images and inclusion of a complementary iAF markers.

Figure 1

• SA- galactosidase staining images show only one area of the plate. It is important to show if all the analyzed samples/plates are positive to galactosidase at the same levels.

• The number of cell/well (Figure 1B) do not correlate with the % of the Ki67 positive cells. The trend of %Ki67 shows decreasing in cell proliferation with time (72hrs) in all the conditions. Representative images do not show a clear Ki67 staining to demonstrate differences/similarities between groups. Are the galactosidase and Ki67 positive cells distinct populations?

Figure 2

• Fold change in the SOD1 SOD2 and CAT, NOX2 and NOX4 gene are slightly different between the experimental and control treatments in NP vs iAF. Then, it is difficult attribute differences in the ROS response of the NP and iAF

Figure 3

• Line 360. Figure legend should be Figure 3 instead of Figure1

• Images of P16 and P21 immunostaining do not show positive staining or is difficult to see them. I strongly suggest images at 10x and 20x magnification then H2O2 changes could be visualized.

• Line 378. Figure legend should be Figure 4 instead of Figure2.

• Cell quantification of Col1, Col2 and Aggrecan immunostaining and images do not show the same data. It is not clear what is the staining vs background levels. Immunostaining must reflect the quantification data from the graphs

Figure 5

• Line 399. Figure legend should be Figure 5 instead of Figure3.

• I highly recommend have a live/dead assay as support data to be certain about the percentage of live cells and the successful 3D cultures.

Figure 6

• Line 421. Figure legend should be Figure 6 instead of Figure4.

• Line 427. Figure legend p21 instead of p12

6. PLOS authors have the option to publish the peer review history of their article (what does this mean?). If published, this will include your full peer review and any attached files.

Reviewer #1: No

Reviewer #2: No

---

## [Author Response · Author response to Decision Letter 0]

13 Jun 2023

PONE-D-22-34265

Senescent response in inner annulus fibrosus cells in response to TNFα, H2O2, and the nucleus pulposus SASP secretome

Reviewer #1: 

The study “Senescent response in inner annulus fibrosus cells in response to TNFα, H2O2, and TNFα-induced nucleus pulposus senescent secretome” is evaluating an interesting and clinically relevant topic. It is proposing a novel difference between NO and iAF cells, but the data is not quite there to support the theory.

General comments:

Comment 1-1

- The paper is aiming to compare NP and iAF cells but both cell types are often not included in the same set of experiments. Please correct.

Authors’ Action 

Thank you for your comment. We apologize for the confusion. Actually, the hypothesis of the study was not to compare iAF and NP cells, but rather to evaluate iAF cells under conditions which we know should induce senescence as we have shown for nucleus pulposus cells (Ashraf et al. 2021 doi: 10.1096/fj.202002201R). This former paper describes TNFα induced senescence in NP cells and explores mechanisms. Thus, we did not include those experiments/results, but instead we provided the reference of this previously published paper in the revised version of this manuscript. 

We do acknowledge that we made some comparisons between the response of NP and iAF cells to TNFα, however, this was only to provide the support for our rationale for exploring a potential mechanism for this differential senescence sensitivity and also to provide context for the role of ROS/H2O2 regulation within the iAF. We have modified the abstract to state this. Also we have re-ordered these descriptions and modified the results on page 16, starting on line 327 which we hope makes it clearer that these comparisons to NP are provided only to confirm they are responding differently (and also to serve as a positive control to demonstrate the reagents were effective).

However, we did repeat the SA-βGal and p16 staining experiments using NP and iAF cells obtained from the same disc harvests as requested. The staining was done at the same time and show similar results to the original experiments (data shown Supplemental Figure 3). 

Comment 1-2

-Please change the title to reflect the study.

Authors’ Action 

 We hope that with the rewrite of the results that it is clearer that the NP studies are present to only support the differential response of the iAF. The original title focuses on the study which is the characterization of iAF cells – the main aim of the study. As some of the NP results are in another paper, we would prefer to keep the title the same (Senescent response in inner annulus fibrosus cells in response to TNFα, H2O2, and TNFα – induced nucleus pulposus cell senescent secretome). However, we are willing to change it if the reviewers feel it is better to include NP in the title.

Comment 1-3

-The references are outdated and don’t reflect the current state of the art. Please update.

Authors’ Action 

Thank you for your comment, we apologize for this. Where appropriate older references have been replaced by more recent publications and the reference list revised to incorporate them and deletion of older papers. 

Comment 1-4

-A positive control for each AB should be included in each experiment for confirmation.

Authors’ Action

Positive immunostaining control images have been added as supplemental figure 4 and referenced on Page 16, line 3-4 and page 20, line 433. 

 A legend was added that describes this as follows: “Immunohistochemical positive control images of (A) NP cells stained with antibody reactive with p16 following 24hr exposure to TNFα. (B-C) Tissue formed by nucleus pulposus cells grown for 2 weeks in 3D culture stained for ACAN (B) and COL2 (C). (D) Native bovine IVD tissue stained with antibody reactive with COL1. DAPI was used as a nuclear counterstain.” 

Comment 1-5

-In general, the IHC are of low quality and are not reflecting the quantifications. Please correct.

Authors’ Action 

Control matched linear adjustment of brightness and contrast was made to figure images, where necessary, to improve visibility of scaled down immunofluorescent images. We hope you can see them more readily.

Comment 1-6

-The figure legends in the text are mislabeled. For example, there are 3 figure legends labelled Fig 1. Please correct.

Authors’ Action

We apologize for this error and the legends have been renumbered to correct this. We realize that this is occurring during the pdf conversion of the document and we are unable to prevent the renumbering. We have asked for help from the editorial office and hope you will receive the correct version of the revised manuscript.

Comment 1-7

-The supplementary figure S2 is not mentioned in the text.

Authors’ Action

Thank you for picking this up. We have now referenced Supplemental Figure 2 in the text to highlight that iAF cells treated with TNFα or H2O2 did not increase cell death or apoptosis and that iAF or NP cells treated with DPI did not increase cell death. (Page 17, lines 348-350 and page 19, line 398).

Comment 1-8

-The number of biological replicates seems too low especially since there is significant variability between the 3 biological replicates.

Authors’ Action 

All independent experiments were performed at least three times (3 biological replicates) using at least three different caudal spines and each independent experiment had three technical replicates unless otherwise specified. We believe in the original figures, the variability between the biological replicates may have looked exaggerated because the y-axis range was too small. We have corrected the y-axis range in some of the figures to standardize the y-axis range across experiments where it was appropriate and did not cause loss of information. We have also included an acknowledgement of the possible sample size limitation in the discussion. (Page 28, lines 607-609)

Specific questions,

Comment 1-9

Figure 1. It is clear from the literature that SA-βGal is overestimating and p16 underestimates the number of senescent cells, but I have never seen a disconnect of this magnitude. It would help interpret the data if NP and iAF cells were treated and analyzed simultaneously.

Authors’ Action

Thank you for your comment. As we described in our response to comment 1-1,

we repeated the SA-βGal and p16 staining experiments on NP and iAF cells from the same population (same cell isolation) at the same time. Our results remain consistent to the original experiments. This data is show in Supplemental Figure 3. A legend was also added in on page 55.

Comment 1-10

Figure 1, B, please provide the same analysis for NP cells. The cells that escaped senescence would continue to proliferate and comparing the two cell types would indicate if the presence of senescent cells affects the continued proliferation.

Authors Action 1-10

Thank you for your comment. As mentioned above, the aim of the study was not to compare iAF and NP cells, but rather to evaluate the conditions in which iAF cells undergo senescence. We hope with the changes to the text of the manuscript this is now clearer.

Evaluating percent Ki67 positive cells was done as a second way to confirm that TNFα was not inducing growth arrest in iAF cells, which is used as another marker of senescence. Quantifying Ki67 cells has been shown to be a good way to assess this (Lawless C. et al. 2010 doi: 10.1016/j.exger.2010.01.018). The conclusion was supported by doing a growth curve to show cell number is increasing in these cells. If senescence was occurring, a reduction in the number of cells relative to untreated control would be expected if a substantial fraction of the total cell population underwent senescence. The positive control (serum starvation) showed growth arrest can be detected with this approach.

Comment 1-11

E, F, there are no visible p16-positive cells, how can about 10% detected? Please provide images that reflect the quantification.

Authors’ Action

We apologize for this. Control matched linear adjustment of brightness and contrast was made to figure images, where necessary, to improve visibility of scaled down immunofluorescent images.

Comment 1-12

Figure 2. The y-axis should have the same scale across the analysis to allow for comparison.

B and E. The iAF data is much more variable than NP and would need more biological repetitions to give solid data. 

Authors’ Action 

As we described in our response to 1-8, we have adjusted the y-axis across all of the graphs in the study to be consistent when comparing the same gene/assay. As mentioned above, we have also included an acknowledgment of the sample size limitation in the discussion. (Page 28, lines 607-609).

Comment 1-13

I detail but it seems that the image in 2D is not the same for Dapi and CellROX

Authors’ Action 

Thank you for your comment. We apologize for the confusion. The image in figure 2D is the same, however, it was mistakenly flipped. We have corrected this in the revised manuscript. 

Comment 1-14

G-H The data is too variable to make the proposed conclusions. PCR data usually requires a minimum of n=6 to give solid data. Was a power analysis performed? I would also suggest presenting the data normalized to its own untreated control (2–∆∆Ct). Please mention the ct values of the housekeeping gene under the different treatments. Please add more biological replicates.

Authors’ Action 

Thank you for your comment. In this study we did not use a power analysis as most gene expression data was evaluated using at least 3 independent biological replicates, each having 3 technical replicates (total of 9 samples per experimental condition). We have used this in our previous studies and this number was sufficient to detect differences. However we have added this in as a potential limitation of our study in the discussion on (Page 28, lines 607).

With regards to the 2–∆∆Ct, we chose to use 2-∆CT over 2-∆∆CT as we believe this more accurately reflects variance within the control and treatment conditions. When using 2-∆∆CT all control samples are set to 1 which makes the standard deviation of the control condition 0. We think this can be misleading when examining only 2 conditions. 

We have provided the average±SD of 18s rRNA Ct values in the figure legends where appropriate. (Page 17, lines 359-362; page 19, lines 409-410; page 20, lines 424-429; page 22, line 469) 

Comment 1-15

Figure 3. Please show data for NP and iAF cells to facilitate comparison and validate the conclusion.

Authors’ Action 

The main aim of this figure was to evaluate the senescence response of iAF cells exposed to H2O2 so adding in NP data does not advance this. As described in the response to comment 1-1 we were not evaluating NP response to H2O2. We hope with the modifications to the manuscript described above that this is now clear. While future work could explore the differential response in greater detail, we first looked to characterize the iAF response to physiologic stimuli thought to contribute to senescence within the IVD. 

Comment 1-16

B-C please show images reflective of the quantification.

Authors’ Action 

We apologize for this. Control matched linear adjustment of brightness and contrast was made to figure images, where necessary, to improve visibility of scaled down immunofluorescent images.

Comment 1-17

Figure 4A, Please see comments in 2G-H

Authors’ Action 

Please see authors’ action 1-14. 

Comment 1-18

Figure 5. Please show data for NP and iAF cells to facilitate comparison and validate the conclusion.

Authors’ Action 

Thank you for your comment. As mentioned above, the aim of the study was not to directly compare iAF and NP cells. We have previously published on NP cell senescence and the impact on tissue formation (Ashraf et al. 2021 doi: 10.1096/fj.202002201R and Ashraf S et al 2020 doi: 10.1002/jor.24476). 

Comment 1-19

Please adjust the discussion in light of the additional experiments.

Authors Action 1-19

Thank you for your comment. We have made changes to the discussion to present some limitations that were highlighted by the reviewers. Although additional experiments were done and added to the supplemental figures, they did not substantially change the conclusions.

Reviewer #2: 

Comment 2-1

My suggestions including a detail timeline of the NP and iAF cultures, iAF and NP cell characterization at the expression level, curves of Dose- effect of H2O2 and TNF specific for iAF and NP, time effect to recognize the ability of the iAF and NP cells to respond to the stressors (TNFα, H2O2). 

Authors’ Action 

Thank you for your comment. The dose of TNFα was selected based on our previous work which showed that NP cells underwent senescence at that concentration (40ng/mL). A time course was not done in this study as we wanted to determine if iAF cells responded similarly to TNFα at a concentration known to induce senescence in other cell types. We have added this to the text to make this clearer on page 6, lines 115-116.

Comment 2-2

Unfortunately, immunostaining and in general images doesn’t reflect the quantification results. I strongly suggest high magnification in the images and inclusion of a complementary iAF markers.

Authors’ Action 

We apologize for this. Control matched linear adjustment of brightness and contrast was made to figure images, where necessary, to improve visibility of scaled down immunofluorescent images. 

Unfortunately, there is no marker specific to iAF cells which is why we are unable to do this evaluation. iAF cells express both collagen type II and type I, whereas outer AF expresses only type I collagen. We use the collagen types as surrogate biomarkers for iAF cells.

Comment 2-3

Figure 1 

• SA- galactosidase staining images show only one area of the plate. It is important to show if all the analyzed samples/plates are positive to galactosidase at the same levels.

Authors’ Action 

Thank you for your comment. We have included a low magnification image of the entire culture stained with SA-βGal (Figure 1A). Although we show a representative high power image, it should be noted, we used a standardized method of counting cells in all our cultures for SA-βGal that eliminates selection bias of areas to count. This methodology is described in detail in Supplementary figure 5. We modified the description in the methods section on page 11, lines 222-223 as follows:

“The images were captured from the same location in each well using a standardized well-overlay method (described in S5 methods).”

Comment 2-4

• The number of cell/well (Figure 1B) do not correlate with the % of the Ki67 positive cells. The trend of %Ki67 shows decreasing in cell proliferation with time (72hrs) in all the conditions. Representative images do not show a clear Ki67 staining to demonstrate differences/similarities between groups. Are the galactosidase and Ki67 positive cells distinct populations?

Authors’ Action 

Thank you for your comment. Ki67 stains proliferating cells from G1 through M. We likely have cells in all stages of the cell cycle so the percentage of Ki 67 positive cells may not exactly correlate with the cell number observed in the growth curve. Secondly, Ki67 does not stain all proliferating cells equally, as protein levels can drop during G0/1 for example. Thus, the levels of Ki67 can be heterogeneous (Miller I, et al. 2018 doi: 10.1016/j.celrep.2018.06.110.). The authors also show “Ki67 levels fall as a function of time since quiescence entry, quiescent cells re-entering the cell cycle will have varying levels of Ki67 depending on how long the cells have been quiescent”. The positivity threshold we used to determine a positive (vs negative) stained cell may have decreased the sensitivity of this evaluation and under-represented total proliferating cells. However, this threshold was the same for all treatment conditions evaluated. We would argue that comparing percent Ki67 positivity and growth curves directly may not be appropriate given the experimental and biological constraints. 

With regards to the decreasing Ki67 positivity over time, this was expected because we did not change the media over the 72hrs following TNFα removal. This was done as previous studies have shown that the secretome of senescence cells can signal in an autocrine and paracrine manner to induce further senescent changes. In this way, we hypothesized not changing the media would give the greatest likelihood of observing senescence effects in the TNFα treated iAF cells – which we still did not observe. We have now added in a statement stating that the media was not changed during the evaluation of the growth curves to clarify this on page 10, line 209-210.

With regards to distinct cell populations, cells used for the Ki67 evaluation are fixed differently (paraformaldehyde) than cells stained for SA-βGal (2% formaldehyde/0.2% glutaraldehyde) so it is not possible to co-stain cells and determine if these positive cells are distinct populations. To try to address this we did a growth curve to show that cell number was increasing which supports the finding that TNFα treatment is not inducing senescence in iAF cells. Although this does not exactly address your question it does provide additional information and is used by others to evaluate for the presence/absence of senescence (Gonzalez-Gualda E. et al. 2021 doi: 10.1111/febs.15570). 

Comment 2-5

Figure 2

• Fold change in the SOD1 SOD2 and CAT, NOX2 and NOX4 gene are slightly different between the experimental and control treatments in NP vs iAF. Then, it is difficult attribute differences in the ROS response of the NP and Iaf

Authors’ Action 

Thank you for your comment. The data is not expressed as fold change. It is expressed as 2-∆CT as we believe this better shows the variances within the control and treatment conditions. When using 2-∆∆CT control samples are set to 1 which makes the standard deviation of the control condition 0. This can make fold change data less suitable for statistical evaluation in a two condition experiment.

With respect to the differential expression we observe between the NP and iAF cells in the untreated control condition, this was expected. NP and iAF are distinct cell populations derived from different embryonic lineages with unique transcriptomes. Previous studies have highlighted a differential expression of genes between untreated iAF and NP cells such as for SOD2 [Kudelko, M et al. 2021 doi: 10.1016/j.mbplus.2021.100082, Panebianco, CJ et al. 2021 doi: 10.1096/fj.202101149R]. As untreated iAF and NP cells had differential gene expression, that is why we analyzed their respective transcriptional response to TNFα independently and comparison was to their respective control cells only. Thus, it is possible to detect differences in ROS response in NP or iAF cells.

Comment 2-6

Figure 3

• Line 360. Figure legend should be Figure 3 instead of Figure1

Authors’ Action 

Thank you for picking this up and apologize for the typo. This has been corrected in the revised version of the manuscript. We realize that this is occurring during the pdf conversion of the document and we are unable to prevent the renumbering. We have asked for help from the editorial office and hope you will receive the correct version of the revised manuscript.

Comment 2-7

• Images of P16 and P21 immunostaining do not show positive staining or is difficult to see them. I strongly suggest images at 10x and 20x magnification then H2O2 changes could be visualized.

Authors’ Action 

We apologize for this. Control matched linear adjustment of brightness and contrast was made to figure images, where necessary, to improve visibility of scaled down immunofluorescent images. We hope with this change, immunostaining can be seen.

Comment 2-8

• Line 378. Figure legend should be Figure 4 instead of Figure2.

Authors’ Action 

Thank you for picking this up and apologize for the typo. This has been corrected in the revised version of the manuscript. We realize that this is occurring during the pdf conversion of the document and we are unable to prevent the renumbering. We have asked for help from the editorial office and hope you will receive the correct version of the revised manuscript.

Comment 2-9

• Cell quantification of Col1, Col2 and Aggrecan immunostaining and images do not show the same data. It is not clear what is the staining vs background levels. Immunostaining must reflect the quantification data from the graphs (Figure 3)

Authors’ Action 

Thank you for your comment. We have increased the brightness and contrast of these images to improve visibility in the revised manuscript. We think that the images and quantification do correlate, as the % positive cells shown in the images fall within the range displayed in the graphical data. For example, in the quantification of COL2 (Fig4B) the control sample in the image displayed has 45.4% positive cells, where the TNFα treated condition has 21.2% positive cells. This is within the range of plotted results on the associated graph (Control: 33-46%, TNFα: 15-25%). 

Comment 2-10

• Line 399. Figure legend should be Figure 5 instead of Figure3.

Authors’ Action 

Thank you for picking this up and apologize for the typo. This has been corrected in the revised version of the manuscript. We realize that this is occurring during the pdf conversion of the document and we are unable to prevent the renumbering. We have asked for help from the editorial office and hope you will receive the correct version of the revised manuscript.

Comment 2-11

• I highly recommend have a live/dead assay as support data to be certain about the percentage of live cells and the successful 3D cultures. 

Authors’ Action 

Thank you for your comment. We had performed Live/Dead assays and TUNEL staining on TNFα and H2O2 treated iAF cells (Supplemental figure 2). In these studies, none of the treatment conditions were found to be cytotoxic. We did not include in-text citations for this figure of the original manuscript which we corrected in the revised version of the manuscript. (Page 17, lines 348-350 and Page 19, lines 396-398).

Comment 2-12

• Line 421. Figure legend should be Figure 6 instead of Figure 4.

Authors Action 

Thank you for picking this up and apologize for the typo. This has been corrected in the revised version of the manuscript. 

Comment 2-13

• Line 427. Figure legend p21 instead of p12

Authors Action 2-13

Thank you for picking this up and apologize for the typo. This has been corrected in the revised version of the manuscript (now on page 19, line 408).

---

## [Decision Letter · Decision Letter 1]

11 Jul 2023

PONE-D-22-34265R1Senescent response in inner annulus fibrosus cells in response to TNFα, H2O2, and TNFα-induced nucleus pulposus senescent secretomePLOS ONE

Dear Dr. Kandel,

Thank you for submitting your manuscript to PLOS ONE. After careful consideration, we feel that it has merit but does not fully meet PLOS ONE’s publication criteria as it currently stands. Therefore, we invite you to submit a revised version of the manuscript that addresses the points raised during the review process.

As Reviewer 1 pointed out in their first review, a n = 3 might be too low to draw solid conclusions about the effect of treatments, especially when considering the minor (while significant) changes between groups. Please acknowledge this limitation in greater detail in the discussion. In addition, as reviewer 1 requests, please add data about TNFaR expression to the manuscript. 

Regarding Figure 6: why has 6 B 6 data points for quantification? The figure legend states that it was n=3. Was the statistics performed with n=3 or n=6? Please correct the figure and perform statistics for n=3.

We look forward to receiving your revised manuscript.

Kind regards,

Svenja Illien-Jünger, Ph.D.

Academic Editor

PLOS ONE

Reviewers' comments:

Reviewer's Responses to Questions

**Comments to the Author**

1. If the authors have adequately addressed your comments raised in a previous round of review and you feel that this manuscript is now acceptable for publication, you may indicate that here to bypass the “Comments to the Author” section, enter your conflict of interest statement in the “Confidential to Editor” section, and submit your "Accept" recommendation.

Reviewer #1: (No Response)

Reviewer #2: All comments have been addressed

2. Is the manuscript technically sound, and do the data support the conclusions?

Reviewer #1: Partly

Reviewer #2: Yes

3. Has the statistical analysis been performed appropriately and rigorously? 

Reviewer #1: Yes

Reviewer #2: Yes

4. Have the authors made all data underlying the findings in their manuscript fully available?

Reviewer #1: (No Response)

Reviewer #2: Yes

5. Is the manuscript presented in an intelligible fashion and written in standard English?

Reviewer #1: Yes

Reviewer #2: Yes

6. Review Comments to the Author

Reviewer #1: The manuscript is significantly improved but a few concerns remain.

Dr. Lemaitre showed in 2007 that cells in non-degenerate IVD tissue have very low TNFAR expression (Arthritis Research & Therapy 2007, 9:R77 (doi:10.1186/ar2275). Based on this it would be important to determine if bovine NP and iAF cells express the same level of the receptor. This knowledge would confirm if the results were due to the availability of the receptor or if the downstream signalling differs. Please add data and discussion.

Figure 3 E control is flipped.

Reviewer #2: The study Senescent response in inner annulus fibrosus cells in response to TNFa, H202, and TNFa-induced nucleus pulposus senescent secretome. The study proposes NP and inner AF differences particularly in the way to respond to TNFa, H2O2 and in the cell coculture by influence on NP-TNF treated cells. The proposing idea is novel and interesting for the Disc community. My questions-concerns were address in the proper way, however the manuscript still require a little bit of improvement at the edition level.

Line 1-37. I highly recommend include the socioeconomic impact of the back pain and DDD at the worldwide, because this is one of the most prevalent issues at global level.

46-47 and 81-82. Sentence “many to believe” Is better to write a fact than write an general opinion.

Methods section

Be homogeneous in the way of including the company, country of origin and use reagents, e.g. sometimes is general: line 94, “Winset, 318-010-cl” ,line 101 “Winset products” sometimes is very specific: line 95 “protease (Type XIV, P5146 Sigma-Aldrich, St. Louis, MO, USA), sometimes even is not included the company name e.g. FBS, line 104. Please correct in all methodology.

Line 99: even if the digestion timing was variable is not the same 16 hrs than 10hrs or 20hrs so please be most precise with the incubation times, if someone try to reproduce your experiments, issues as over digestion or poor tissue dissociation can affect enormously the results.

Line 102, 105, 196, 221, 238, 249: same situation than before. I assume that cell counting was performed by manual or automated counting so by precisely with the number of cells.

Line 145: Formation of the tissue sheet. It was not described what kind of supportive material was used to seed the iAF . Authors can be precise if was home made or sheets were from specific company and how the 3D structures were performed.

Line 296: Figure 1. Unfortunately, it was not uploaded the new Figure 1 instead is the figure 2 in the attachments. In the pdf the resolution is very poor, I would like to the high quality image.

Line 329-330: Figure 2: Although in the text it was mentioned the AF and NP comparison. Is difficult to interpret the figure. Is unclear which data/graphs correspond to iAF and wich to NP. Maybe subtitles or frames-grouping can help to understand better the figure.

Figure 3,4: Same than above make clear NP and iAF graphs.

S4: Figure S4 need the figure legend or subtitles on top of images. They are only label with A,B,C,D.

Discussion section: The discussion section was deep enough, covering different perspectives of the topic and the body of information is sufficient to support the data depicted in the manuscript.

7. PLOS authors have the option to publish the peer review history of their article (what does this mean?). If published, this will include your full peer review and any attached files.

Reviewer #1: No

Reviewer #2: No

---

## [Author Response · Author response to Decision Letter 1]

28 Aug 2023

Academic Editor:

1) As Reviewer 1 pointed out in their first review, a n = 3 might be too low to draw solid conclusions about the effect of treatments, especially when considering the minor (while significant) changes between groups. Please acknowledge this limitation in greater detail in the discussion. 

- Thank you for your comment. We have expanded our discussion of this limitation including “Given that some of the effect sizes, although significant, were limited, the biologic relevance of these findings will require further investigation.” (lines 604-605)

2) In addition, as reviewer 1 requests, please add data about TNFaR expression to the manuscript. 

- Thank you for your comment. We have analyzed for TNFR1 gene and protein expression in iAF and NP. The data is added into the manuscript as supplemental figure 5. We have also described the findings in the abstract and results “Additionally, gene and protein expression show that TNFR1 is expressed by both NP and iAF cells, however, TNFR1 protein expression is significantly higher in iAF cells than NP cells (Fig. S5).” (lines 368-370). The discussion was modified to incorporate this information. For example we added to the discussion “Furthermore, both iAF and NP cells had measurable levels of TNFR1 gene and protein expression, which suggests the differential senescence and degeneration response to TNFα is likely downstream of the TNFR1.” (lines 561-563). We have included the details of the antibodies and primers used in tables 1 - 3. As well the western blot method and densitometry description has been added into the supplemental methods figure now labelled Supplemental Figure 6 Methods.

3)Regarding Figure 6: why has 6 B 6 data points for quantification? The figure legend states that it was n=3. Was the statistics performed with n=3 or n=6? Please correct the figure and perform statistics for n=3.

- We apologize for this error. We have clarified in the legend of figure 6 that N=6 for collagen and aggrecan immunostaining (line 477). The statistics were all done with the appropriate replicates. 

Reviewer #1: 

The manuscript is significantly improved but a few concerns remain.

Comment 1-1

Dr. Lemaitre showed in 2007 that cells in non-degenerate IVD tissue have very low TNFAR expression (Arthritis Research & Therapy 2007, 9:R77 (doi:10.1186/ar2275). Based on this it would be important to determine if bovine NP and iAF cells express the same level of the receptor. This knowledge would confirm if the results were due to the availability of the receptor or if the downstream signalling differs. Please add data and discussion.

- Thank you for your comment. We have added TNFR1 gene and protein expression data to supplemental figure 5 and have commented on these data in the results and discussion (see academic editor comment 2). 

Interestingly, iAF cells had higher protein expression of TNFR1 than NP cells (despite similar gene expression). This suggests either that the differential response in iAF vs NP cells to TNFα is due to (1) the differential TNFR1 concentrations induce a differential response (although we believe this is unlikely given NP cells undergo senescence when exposed to TNFα despite low TNFR1 protein expression), (2) that the differential response is due to downstream factors that influence TNFR1 response(s), or (3) that the differential response may be due to activation of other TNFα superfamily receptors (a family that comprises 27 different receptors), or possibly (4) a combination of the above. 

Comment 1-2

Figure 3 E control is flipped.

- Thank you for detecting this. We have addressed this issue in the revised version of the manuscript. 

Reviewer #2: 

The study Senescent response in inner annulus fibrosus cells in response to TNFa, H202, and TNFa-induced nucleus pulposus senescent secretome. The study proposes NP and inner AF differences particularly in the way to respond to TNFa, H2O2 and in the cell coculture by influence on NP-TNF treated cells. The proposing idea is novel and interesting for the Disc community. My questions-concerns were address in the proper way, however the manuscript still require a little bit of improvement at the edition level.

Comment 2-1

Line 1-37. I highly recommend include the socioeconomic impact of the back pain and DDD at the worldwide, because this is one of the most prevalent issues at global level.

- Thank you for your comment. We have added a statement to the introduction discussing the global impact of IVD degeneration/back pain beyond just Canada. (lines 40-41). A reference was added to support this statement and all the subsequent references were renumbered.

Comment 2-2

46-47 and 81-82. Sentence “many to believe” Is better to write a fact than write an general opinion.

- Thank you for your comment. We have changed the wording where applicable. 

Comment 2-3

Methods section

Be homogeneous in the way of including the company, country of origin and use reagents, e.g. sometimes is general: line 94, “Winset, 318-010-cl” ,line 101 “Winset products” sometimes is very specific: line 95 “protease (Type XIV, P5146 Sigma-Aldrich, St. Louis, MO, USA), sometimes even is not included the company name e.g. FBS, line 104. Please correct in all methodology.

- Thank you for your comment. We have changed all product references to be consistent ("product number, company name, city and country”, ) where applicable. 

Comment 2-4

Line 99: even if the digestion timing was variable is not the same 16 hrs than 10hrs or 20hrs so please be most precise with the incubation times, if someone try to reproduce your experiments, issues as over digestion or poor tissue dissociation can affect enormously the results.

- Thank you for your comment. We have changed the wording in “Cell isolation and monolayer culture to reflect the variance of the digestion time, which was 16hrs ±1hr. (line 107).

Comment 2-5

Line 102, 105, 196, 221, 238, 249: same situation than before. I assume that cell counting was performed by manual or automated counting so by precisely with the number of cells.

- Thank you for your comment. We have added a line in the methods describing the manual cell counting was used (line 128 and 130) and removed “approximately” from all cell counting numbers. 

Comment 2-6

Line 145: Formation of the tissue sheet. It was not described what kind of supportive material was used to seed the iAF. Authors can be precise if was home made or sheets were from specific company and how the 3D structures were performed.

- We apologize for confusion around this method. We formed the cell sheets exactly as described in the methods. Cells were seeded directly on uncoated polystyrene culture plates. Cells were seeded at high density, grown in the complete media described in the manuscript, and cultured for 10 days to accumulate the matrix molecules they produce as extracellular matrix and form a 3D tissue. We have modified the method starting on line 162 to try to clarify this. On lines 173-174 the following statement was added- Under these culture conditions, the cells accumulate extracellular matrix molecules they synthesize and form tissue. 

- However we can see how it can be confusing to say we grew the tissue in 3D when actually it is the high density culture that resulted in tissue formation which is 3D. We have changed this in the results section on lines 437-438 so it now states- cells were also grown in high density culture so they formed tissue.

We also removed 3D and replaced it with tissue where appropriate, eg line 440 3D was changed to tissue.

Comment 2-7

Line 296: Figure 1. Unfortunately, it was not uploaded the new Figure 1 instead is the figure 2 in the attachments. In the pdf the resolution is very poor, I would like to the high quality image.

- We were able to identify why the figures were being mislabeled and this has now been fixed so the appropriate figure 1 is provided. We apologize for the confusion.

Comment 2-8

Line 329-330: Figure 2: Although in the text it was mentioned the AF and NP comparison. Is difficult to interpret the figure. Is unclear which data/graphs correspond to iAF and which to NP. Maybe subtitles or frames-grouping can help to understand better the figure.

- Thank you for your comment. We have added additional labels to subfigures 2 G and H to help distinguish iAF vs NP data. Subfigures I and J have titles above the graphs to highlight NP vs iAF. 

Comment 2-9

Figure 3,4: Same than above make clear NP and iAF graphs.

- Figure 3 and 4 only present iAF data. 

Comment 2-10

S4: Figure S4 need the figure legend or subtitles on top of images. They are only label with A,B,C,D.

- Thank you for your comment. We have included labels to supplemental figure 4 indicating the tissue used and the antibody used. 

Comment 2-11

Discussion section: The discussion section was deep enough, covering different perspectives of the topic and the body of information is sufficient to support the data depicted in the manuscript.

- Thank you for all of your comments.

---

## [Decision Letter · Decision Letter 2]

19 Sep 2023

PONE-D-22-34265R2Senescent response in inner annulus fibrosus cells in response to TNFα, H2O2, and TNFα-induced nucleus pulposus senescent secretomePLOS ONE

Dear Dr. Kandel,

Thank you for submitting your manuscript to PLOS ONE. After careful consideration, we feel that the manuscript has greatly improved and only needs a minor revision before it can be accepted. Therefore, we invite you to submit a revised version of the manuscript that addresses the points raised during the review process.

We look forward to receiving your revised manuscript.

Kind regards,

Svenja Illien-Jünger, Ph.D.

Academic Editor

PLOS ONE

Journal Requirements:

Reviewers' comments:

Reviewer's Responses to Questions

**Comments to the Author**

1. If the authors have adequately addressed your comments raised in a previous round of review and you feel that this manuscript is now acceptable for publication, you may indicate that here to bypass the “Comments to the Author” section, enter your conflict of interest statement in the “Confidential to Editor” section, and submit your "Accept" recommendation.

Reviewer #1: (No Response)

Reviewer #2: All comments have been addressed

2. Is the manuscript technically sound, and do the data support the conclusions?

Reviewer #1: Partly

Reviewer #2: Yes

3. Has the statistical analysis been performed appropriately and rigorously? 

Reviewer #1: I Don't Know

Reviewer #2: Yes

4. Have the authors made all data underlying the findings in their manuscript fully available?

Reviewer #1: Yes

Reviewer #2: Yes

5. Is the manuscript presented in an intelligible fashion and written in standard English?

Reviewer #1: Yes

Reviewer #2: Yes

6. Review Comments to the Author

Reviewer #1: Thanks for providing additional data. It is very clear from the additional data that TNF receptor expression at the protein level is many times lower in NP than AF cells. This is not clearly acknowledged. The response to TNF will likely be much milder in NP cells (and potentially resulting in a different secretory phenotype) which would explain why senescence is not induced to the same extent as in AF cells. Activating the cells by for example IL1 might equally induce senescence in both cell types as is shown for induction with peroxide. The difference is therefore likely based on receptor availability rather than different responses. Unless the secretory phenotype is verified at the protein level together with an evaluation of the downstream signalling, it must be clearly stated in the discussion and abstract that the difference may be due to receptor availability which could result in differences (or a much weaker response) of the secretory phenotype. Figure S5 is critical for the discussion and should included in the main text of the manuscript and not as a supplement figure.

Reviewer #2: Authors properly addressed the previous comments. A significant improvement of the manuscript was done. I do not have further comments.

7. PLOS authors have the option to publish the peer review history of their article (what does this mean?). If published, this will include your full peer review and any attached files.

Reviewer #1: No

Reviewer #2: **Yes: **Martha Elena Diaz-Hernandez

---

## [Author Response · Author response to Decision Letter 2]

12 Oct 2023

PONE-D-22-34265R2

Senescent response in inner annulus fibrosus cells in response to TNFα, H2O2, and TNFα-induced nucleus pulposus senescent secretome.

PLOS ONE

Dear Dr. Illien-Jünger,

Thank you for reviewing our manuscript (PONE-D-22-34265R2) and allowing us the opportunity to respond to the reviewer’s comments. I will note we have made some grammatical changes that we identified as we re-read the manuscript. These are tracked as well.

Reviewer #1: 

Thanks for providing additional data. It is very clear from the additional data that TNF receptor expression at the protein level is many times lower in NP than AF cells. This is not clearly acknowledged. The response to TNF will likely be much milder in NP cells (and potentially resulting in a different secretory phenotype) which would explain why senescence is not induced to the same extent as in AF cells. Activating the cells by for example IL1 might equally induce senescence in both cell types as is shown for induction with peroxide. The difference is therefore likely based on receptor availability rather than different responses. Unless the secretory phenotype is verified at the protein level together with an evaluation of the downstream signalling, it must be clearly stated in the discussion and abstract that the difference may be due to receptor availability which could result in differences (or a much weaker response) of the secretory phenotype. Figure S5 is critical for the discussion and should included in the main text of the manuscript and not as a supplement figure.

Author’s Response:

We appreciate your comments. We acknowledge the substantial difference in TNF-α receptor expression at the protein level between NP and AF cells, which may indeed underlie the variations in senescence induction and the secretory phenotype observed between the two cell types. 

We have addressed this point in the results section, emphasizing the magnitude of this difference (Lines 373-377). We have moved Figure S5 into the text and is now Figure 3. The subsequent figures have been renumbered and modified in the manuscript as appropriate. 

We have added into the discussion the concept of receptor availability possibly driving the observed differences, and also included a statement about the need for further experiments to verify these hypotheses at the downstream signaling ( Lines 534-536; Lines 560- 563) 

We modified the Abstract to reflect the presence of TNFR1 in iAF cells (Line 21-22).

Reviewer #2: 

Authors properly addressed the previous comments. A significant improvement of the manuscript was done. I do not have further comments.

Author’s Response:

Thank you very much for helping us improve our manuscript.

We hope with these changes the manuscript is now acceptable for publication in PLOS ONE. Should you require any further information please do not hesitate to contact me.

Yours Sincerely,

Rita Kandel

---

## [Decision Letter · Decision Letter 3]

13 Oct 2023

PONE-D-22-34265R3Senescent response in inner annulus fibrosus cells in response to TNFα, H2O2, and TNFα-induced nucleus pulposus senescent secretomePLOS ONE

Dear Dr. Kandel,

Thank you for submitting your manuscript to PLOS ONE. After careful consideration, we feel that it has merit but does not fully meet PLOS ONE’s publication criteria as it currently stands. Therefore, we invite you to submit a revised version of the manuscript that addresses the points raised during the review process.

Please follow the reviewer's request and discuss the fact that the TNFa receptor is significantly less present on iAF cells. Please add this to the discussion section. 

We look forward to receiving your revised manuscript.

Kind regards,

Svenja Illien-Jünger, Ph.D.

Academic Editor

PLOS ONE

Journal Requirements:

Additional Editor Comments:

Dear Dr. Kandel,

I hope this message finds you well. please follow the reviewer's request about addressing the fact that iAF cells present significantly less of the TNFα receptor at the cell surface. Please see their comments.

Thank you and have a great weekend,

Svenja Illien-Junger

Reviewers' comments:

Reviewer's Responses to Questions

**Comments to the Author**

1. If the authors have adequately addressed your comments raised in a previous round of review and you feel that this manuscript is now acceptable for publication, you may indicate that here to bypass the “Comments to the Author” section, enter your conflict of interest statement in the “Confidential to Editor” section, and submit your "Accept" recommendation.

Reviewer #1: (No Response)

2. Is the manuscript technically sound, and do the data support the conclusions?

Reviewer #1: Yes

3. Has the statistical analysis been performed appropriately and rigorously? 

Reviewer #1: I Don't Know

4. Have the authors made all data underlying the findings in their manuscript fully available?

Reviewer #1: Yes

5. Is the manuscript presented in an intelligible fashion and written in standard English?

Reviewer #1: Yes

6. Review Comments to the Author

Reviewer #1: The manuscript is significantly improved however the fact that iAF cells present significantly less of the TNFα receptor at the cell surface still is not clearly indicated. I fully agree that the cells respond different to TNF which is interesting, but one would expect that given the lower presence of receptor availability.

Please revise to;

Although TNFα is known to induce senescence in NP cells, this study demonstrates that iAF cells

are resistant to TNFα-induced senescence under the conditions examined. iAF cells express TNFα receptor 1 although at a significantly lower level than NP cells.

Furthermore, both iAF and NP cells had measurable levels of TNFR1 gene and protein expression, however the protein level of the receptor was significantly lower in iAF cells, which suggests the differential senescence and degeneration response in iAF cells.

7. PLOS authors have the option to publish the peer review history of their article (what does this mean?). If published, this will include your full peer review and any attached files.

Reviewer #1: No

---

## [Author Response · Author response to Decision Letter 3]

31 Oct 2023

Thank you very much for considering our manuscript for publication in PLOS One. Thank you for the opportunity to respond to your request for change. We have modified the manuscript according to the Editor’s comments. The changes are tracked in the revised manuscript. 

I would like to emphasize that the reviewer misread the results shown in the western blot. iAF, not NP cells, have higher protein expression of TNFR1. We have rewritten this section in the discussion (starting line 540) to make it clearer. We deleted the original statement (starting line 562). We have added in three new references (numbers 67-69) for the new text and all subsequent references were renumbered.

We hope with these changes the manuscript is now acceptable. Should you have any additional questions, please do not hesitate to contact us.

---

## [Editor Report · Decision Letter 4]

2 Nov 2023

Senescent response in inner annulus fibrosus cells in response to TNFα, H2O2, and TNFα-induced nucleus pulposus senescent secretome

PONE-D-22-34265R4

Dear Dr. Kandel,

We’re pleased to inform you that your manuscript has been judged scientifically suitable for publication and will be formally accepted for publication once it meets all outstanding technical requirements.

Kind regards,

Svenja Illien-Jünger, Ph.D.

Academic Editor

PLOS ONE
---

## [Editor Report · Acceptance letter]

28 Dec 2023

PONE-D-22-34265R4 

PLOS ONE

Dear Dr. Kandel, 

I'm pleased to inform you that your manuscript has been deemed suitable for publication in PLOS ONE. Congratulations! Your manuscript is now being handed over to our production team.

Kind regards, 

on behalf of

Dr. Svenja Illien-Jünger 

Academic Editor

PLOS ONE